# MITIGATING ERROR PROPAGATION IN LOW-RANK APPROXIMATION OF LARGE MODELS VIA DISTRIBUTION-AWARE WHITENING

## ABSTRACT

Low-rank approximation has emerged as a cornerstone technique for model compression and parameter-efficient fine-tuning, enabling substantial reductions in computation and memory without altering model architectures. However, existing approaches often overlook the shifts in feature distributions induced by the approximation process, which can lead to error amplification and unstable inference. We propose a distribution-aware whitening framework that dynamically whitens layer inputs based on the evolving feature distributions, ensuring second-order isotropy of input features. This allows that the discarded components in the low-rank approximation are those with minimal impact on model outputs, thereby minimizing cumulative approximation errors across layers. We theoretically analyze how distribution misalignment leads to error propagation and demonstrate that our approach achieves tighter control over layerwise distortion. Extensive experiments across various large language models demonstrate the superiority of our method in post-training compression. Moreover, our method can also serve as an effective initialization for LoRA-style parameter-efficient fine-tuning. Our findings highlight the importance of considering feature distributions in low-rank approximations, paving the way for reliable and effective model compression strategies.

## 1 INTRODUCTION

Low-rank approximation has emerged as a fundamental technique in modern deep learning, particularly for model compression and parameter-efficient fine-tuning (PEFT). By decomposing weight matrices into low-rank components, these methods substantially reduce the number of trainable or deployable parameters, while maintaining compatibility with the original model architecture. This paradigm has enabled a wide range of scalable applications, from efficient deployment of large language models (LLMs) on resource-constrained devices (Dettmers et al., 2023), to lightweight adaptation across diverse downstream tasks (Hu et al., 2022; Zhang et al., 2023).

Despite their widespread adoption, existing low-rank methods typically focus on minimizing the discrepancy between the original and approximated weights—most commonly through Frobenius norm minimization—without fully accounting for the statistical properties of the input activations (Meng et al., 2024). However, in deep neural networks, the effective impact of approximation is determined not solely by the weights themselves, but also by how they interact with the input feature distribution. When activations are anisotropic, i.e., exhibiting directional variance, low-magnitude singular directions in the weight matrix may align with high-energy input components. Truncating such directions can cause disproportionately large output distortion, violating the assumption that small singular values are always negligible. This effect is further amplified in deep models, where compression is applied recursively across layers and prior layer's approximations can induce significant shifts in activation distributions. Although methods such as ASVD (Yuan et al., 2023) and SVD-LLM (Wang et al., 2025b) attempt to introduce data preprocessing strategies to establish a relationship between singular values and model compression loss, they lack mechanisms to account for the evolving input distribution induced by layerwise approximation, and thus exhibit severe performance degradation under high compression ratios.

To this end, we propose a **distribution-aware low-rank approximation framework** that explicitly incorporates the second-order statistics of each layer's inputs during the compression process. Specifically, we apply a whitening transformation to the input activations to ensure that low-rank projections are performed within an isotropic feature space. Unlike approaches that employ a fixed transformation, our method dynamically captures the evolving feature distributions induced by compression in preceding layers, and re-estimates the input covariance at each layer during the forward pass using the compressed activations it receives. This effectively decouples input anisotropy from the spectral truncation of the weights, allowing the approximation to more accurately reflect the output distortion caused by discarded components. Consequently, our framework establishes a self-consistent, distribution-aware pipeline whereby the compression strategy at each layer adapts to the current feature distribution.

We provide a theoretical analysis that characterizes how distribution misalignment amplifies approximation errors. Experimentally, our method consistently outperforms existing low-rank compression approaches such as SVD-LLM (Wang et al., 2025b), achieving superior results not only in standard post-training compression but also in scenarios involving dynamic rank allocation and lightweight post-compression fine-tuning. Moreover, it can also serve as an effective initialization for LoRA-style parameter-efficient adaptation, surpassing recent alternatives such as PiSSA (Meng et al., 2024). Our main contributions are summarized as follows:

- We propose a dynamic, distribution-aware whitening framework that performs low-rank approximation in an isotropic feature space, enabling both effective SVD-based model compression and improved initialization for LoRA-style PEFT method.

- We provide a theoretical analysis of how low-rank approximation errors propagate across layers, showing that dynamically accounting for evolving feature distribution improves approximation fidelity and mitigates cumulative distortion.

- Extensive experiments across a range of large language models demonstrate the superiority of our approach over existing SVD-based compression methods in multiple settings, including standard post-training compression, lightweight post-compression fine-tuning, and dynamic rank allocation.

## 2 RELATED WORK

### 2.1 LOW-RANK APPROXIMATION IN MODEL COMPRESSION

By decomposing the weight matrices and discarding components associated with small singular values, singular value decomposition (SVD) can significantly reduce the parameter count and computational overhead while preserving the original model architecture, thereby enabling hardware-efficient inference (Kim et al., 2015). Recent works have introduced various enhancements to improve compression quality. FWSVD (Hsu et al., 2022) incorporates Fisher information to better assess parameter importance, enabling a more principled truncation strategy. ASVD (Yuan et al., 2023) introduces activation-aware scaling and adaptively assigns ranks based on layer-wise compression sensitivity, achieving competitive performance without retraining. SVD-LLM (Wang et al., 2025b) proposes static whitening and layerwise closed-form updates, leading to improved trade-offs between compression ratio and perplexity performance. SVD-LLM-V2 (Wang et al., 2025a) and DipSVD (Ding et al., 2025) further highlight that uniform compression ratios overlook the varying redundancy across layers, and advocate for dynamically allocating compression budgets per layer. However, these methods commonly neglect the impact of feature distribution shifts introduced during compression, which can exacerbate error propagation. Therefore, our method introduces a distribution-aware whitening mechanism to mitigate the influence of distribution drift, thereby improving the stability and robustness of low-rank compression.

### 2.2 LOW-RANK APPROXIMATION IN PARAMETER-EFFICIENT FINE-TUNING

Large language models (LLMs) contain hundreds of millions to billions of parameters, posing significant computational and memory challenges when adapting to downstream tasks. To address this, Parameter-Efficient Fine-Tuning (PEFT) methods have been proposed to reduce the number of trainable parameters while maintaining competitive performance with full-model fine-tuning (Hu et al.,

2022; Fu et al., 2023; Houlsby et al., 2019; Zhang et al., 2023; Zhao et al., 2022; Lei et al., 2023). Among these, Low-Rank Adaptation (LoRA) (Hu et al., 2022) has gained widespread adoption due to its architectural simplicity and efficiency.

LoRA injects trainable low-rank matrices into the linear layers of a pre-trained model by expressing the update as $\Delta W = AB$, where $A \in \mathbb{R}^{m \times r}$, $B \in \mathbb{R}^{r \times n}$, and $r \ll \min(m, n)$. The original weight $W$ remains frozen, and only $A$ and $B$ are updated, enabling efficient fine-tuning with minimal memory overhead. Despite its efficiency, LoRA typically initializes $A$ with Kaiming initialization and $B$ with zeros, resulting in update directions that are poorly aligned with the pre-trained model. This can lead to slow convergence or suboptimal performance at small ranks.

To improve over this, Meng et al. (2024) proposed PiSSA, which initializes $A$ and $B$ using the top-$r$ singular vectors of the original weight matrix $W$. The remaining components of $W$ are stored in a residual matrix $W_{\text{res}}$, which is frozen during fine-tuning. Building upon this, our work further proposes utilizing a **distribution-aware whitening** mechanism during low-rank initialization. This mechanism enables adaptation in a subspace aligned with the most significant directions of the original model, leading to faster convergence and improved downstream performance.

## 3 METHODOLOGY

In this section, we propose a distribution-aware low-rank approximation framework that accounts for the layer-wise evolution of feature distributions during the approximation process, thereby mitigating the performance degradation typically induced by low-rank compression.

### 3.1 MOTIVATION

Conventional low-rank approximation seeks to minimize the Frobenius norm between the original and approximated weight matrices:

$$\min_{\hat{W}_\ell : \text{rank}(\hat{W}_\ell) \leq k} \|W_\ell - \hat{W}_\ell\|_F, \tag{1}$$

which does not consider the distribution of the input features $X_{\ell-1}$. However, in neural networks, the approximation error of interest lies in the output feature space: $\|W_\ell X_{\ell-1} - \hat{W}_\ell X_{\ell-1}\|_F^2$. Taking expectation over the feature distribution, this error can be expressed as:

$$\mathbb{E}\left[\|(W_\ell - \hat{W}_\ell)X_{\ell-1}\|_F^2\right] = \text{Tr}\left((W_\ell - \hat{W}_\ell)\Sigma_{\ell-1}(W_\ell - \hat{W}_\ell)^\top\right), \tag{2}$$

where $\Sigma_{\ell-1} = \mathbb{E}[X_{\ell-1}X_{\ell-1}^\top]$ denotes the covariance matrix of input feature. This indicates that the approximation error depends not only on the weight matrix but also on the input distribution. When $X_{\ell-1}$ is anisotropic, i.e., its covariance $\Sigma_{\ell-1}$ has dominant eigen-directions, small singular values of $W_\ell$ may correspond to directions along which the input features exhibit high variance, leading to disproportionately large output errors upon truncation. This challenges the common assumption that small singular values contribute negligibly to the output error.

### 3.2 DISTRIBUTION-AWARE LOW-RANK APPROXIMATION

To this end, we propose a distribution-aware low-rank compression framework that dynamically adapts to the evolving feature distributions induced by layerwise approximation. Motivated by the observation that the approximation error depends on both the weight matrix and the input covariance, our method performs low-rank truncation in a normalized feature space where the anisotropy of activations is mitigated.

Specifically, let $X_{\ell-1} \in \mathbb{R}^{d_{\ell-1} \times n}$ denote the input feature to layer $\ell$, produced by the previous layer's (possibly compressed) transformation: $X_{\ell-1} = \phi(\hat{W}_{\ell-1}X_{\ell-2})$, where $\hat{W}_{\ell-1}$ is the low-rank approximation of $W_{\ell-1}$, and $\phi(\cdot)$ represents the inter-layer operations (e.g., activation, normalization, residual addition). Considering that as compression proceeds layer by layer, the distribution of the input features $X_{\ell-1}$ propagated to the $l$-th layer evolves, we dynamically estimate the empirical second-order statistics of $X_{\ell-1}$ according to the current compression state and perform a whitening transformation to normalize the input distribution:

$$S_\ell = (L_\ell L_\ell^\top)^{-1/2}, \quad L_\ell = \text{cholesky}(X_{\ell-1}X_{\ell-1}^\top + \epsilon \cdot \text{diag}(X_{\ell-1}X_{\ell-1}^\top)), \tag{3}$$

where $\epsilon$ is a small constant for numerical stability, and

$$\tilde{X}_{\ell-1} = S_\ell X_{\ell-1}, \quad \text{so that} \quad \tilde{X}_{\ell-1}\tilde{X}_{\ell-1}^\top \approx I. \tag{4}$$

The weight matrix of the current layer is transformed to operate in the whitened space:

$$\tilde{W}_\ell = W_\ell S_\ell^{-1}, \quad \text{so that} \quad W_\ell X_{\ell-1} = \tilde{W}_\ell \tilde{X}_{\ell-1}. \tag{5}$$

We then perform standard low-rank approximation on $\tilde{W}_\ell$: $\tilde{W}_\ell^{(k)} = \arg\min_{\bar{W}:\text{rank}(\bar{W})\leq k} \|\tilde{W}_\ell - \bar{W}\|_F$, and map the result back to the original space: $\hat{W}_\ell = \tilde{W}_\ell^{(k)} S_\ell$.

Under the assumption of ideal whitening (i.e., $\tilde{X}_{\ell-1}\tilde{X}_{\ell-1}^\top = I$), the output reconstruction error satisfies:

$$\|W_\ell X_{\ell-1} - \hat{W}_\ell X_{\ell-1}\|_F = \|\tilde{W}_\ell - \tilde{W}_\ell^{(k)}\|_F = \left(\sum_{i=k+1}^r \sigma_i^2\right)^{1/2}, \tag{6}$$

where $\sigma_i$ are the singular values of $\tilde{W}_\ell$, and $r = \min(d_\ell, d_{\ell-1})$. This result establishes a direct link between spectral truncation in the whitened space and output fidelity under the actual input distribution. See Appendix C for details.

Importantly, this procedure is repeated for each layer $\ell$, using the actual input $X_{\ell-1}$ generated by the compressed preceding layers. We iteratively re-estimate $S_\ell$ based on the propagated compressed activations to capture the evolving distribution induced by prior approximations. This dynamic re-estimation effectively decouples the approximation error from the anisotropy of the input distribution, making the Frobenius-norm-based truncation more faithful to the impact on the output.

**Model Compression.** We perform singular value decomposition (SVD) on $\tilde{W}_\ell = U_\ell \Sigma_\ell V_\ell^\top$, truncate to rank $k$, and reconstruct:

$$\hat{W}_\ell = \tilde{W}_\ell^{(k)} S_\ell = U_\ell[\text{Trunc}_*(\Sigma_\ell)]V_\ell^\top S_\ell. \tag{7}$$

For efficient deployment, we decompose the compressed weight into two matrices:

$$W_{u,\ell} = U_\ell[\text{Trunc}_*(\Sigma_\ell)]^{1/2}, \quad W_{v,\ell} = [\text{Trunc}_*(\Sigma_\ell)]^{1/2}V_\ell^\top S_\ell, \tag{8}$$

so that $\hat{W}_\ell = W_{u,\ell}W_{v,\ell}$, enabling direct substitution with low-rank linear modules for parameter-efficient model compression.

**Parameter-Efficient Fine-Tuning.** Our framework can also serve as a distribution-aware initialization strategy for low-rank adapters in parameter-efficient fine-tuning (PEFT). Unlike LoRA (Hu et al., 2022), which initializes low-rank adapters with Kaiming initialization, or PiSSA (Meng et al., 2024), which utilizes principal singular directions of the pre-trained weights, our method adapts to the input distribution by performing low-rank approximation in the whitened space. Specifically, we compute SVD on $\tilde{W} = WS^{-1}$, and construct low-rank adapters as $\Delta W = W_u W_v$, where $W_u, W_v$ are derived as Eq.(8). This ensures that the update directions align with the dominant signal directions under the current feature distribution, leading to faster convergence while preserving the low parameter count of LoRA-style adapters.

## 3.3 Error Propagation Under Mismatched Whitening

In this subsection, we analyze how different low-rank approximation approaches affect the propagation of approximation error across layers in deep models. In particular, we contrast three approaches: **Raw approximation**, which directly applies low-rank approximation to the original weight matrix without regard to input distribution; **Static whitening** (Yuan et al., 2023; Wang et al., 2025b), which whitens inputs using a fixed matrix computed prior to compression; **Distribution-aware whitening**, which dynamically adapts the whitening matrix to compressed representations at each layer.

**Raw Approximation Error.** Direct low-rank approximation of $W_\ell$ incurs significant error when the input activations $X_{\ell-1}$ are anisotropic, i.e., concentrated in specific directions. The resulting approximation error at layer $\ell$ is given by:

$$E_\ell^{\text{raw}} = \left\|\left(W_\ell - \hat{W}_\ell^{\text{raw}}\right) X_{\ell-1}\right\|_F, \tag{9}$$

where $\hat{W}_\ell^{\text{raw}}$ is the rank-$k$ approximation of $W_\ell$. Since this ignores the input feature statistics, truncating small singular directions in $W_\ell$ may inadvertently erase high-energy components of $W_\ell X_{\ell-1}$.

**Static Whitening and Whitening Drift.** Static whitening improves alignment by using a fixed matrix $S_\ell^{\mathrm{stat}}$ computed from the original (pre-compression) inputs:

$$\tilde{X}_{\ell-1}^{\mathrm{stat}} = S_\ell^{\mathrm{stat}} X_{\ell-1}, \quad S_\ell^{\mathrm{stat}} = \left(X_{\ell-1} X_{\ell-1}^\top + \epsilon I\right)^{-\frac{1}{2}}. \tag{10}$$

Ideally, this transformation yields whitened inputs with identity covariance. However, after compression is applied to earlier layers, the actual inputs to layer $\ell$ become $\hat{X}_{\ell-1}$ rather than $X_{\ell-1}$, leading to a distribution mismatch:

$$\tilde{X}_{\ell-1} = S_\ell^{\mathrm{stat}} \hat{X}_{\ell-1}, \quad \tilde{X}_{\ell-1}\tilde{X}_{\ell-1}^\top = I + \Delta_\ell. \tag{11}$$

We define the spectral norm of the deviation from identity as the *whitening drift*: $\delta_\ell := \|\Delta_\ell\|_2$, which quantifies the degree to which the whitened inputs deviate from isotropy due to the outdated whitening matrix. This drift can be further decomposed into two components:

$$\delta_\ell \leq \underbrace{\left\|S_\ell^{\mathrm{stat}} \Sigma_{\ell-1} (S_\ell^{\mathrm{stat}})^\top - I\right\|_2}_{\text{initial whitening residual}} + \underbrace{\left\|S_\ell^{\mathrm{stat}} (\hat{\Sigma}_{\ell-1} - \Sigma_{\ell-1})(S_\ell^{\mathrm{stat}})^\top\right\|_2}_{\text{compression-induced distribution shift}}, \tag{12}$$

where $\Sigma_{\ell-1}$ and $\hat{\Sigma}_{\ell-1}$ are the input covariance matrices before and after compression, respectively. The first term arises from the numerical-stability regularization ($\epsilon I$) in the whitening process; its magnitude is typically small and can be ignored in practice. The second term captures the distribution mismatch induced by upstream compression. The resulting compression error is bounded by:

$$E_\ell^{\mathrm{stat}} = \left\|\left(\tilde{W}_\ell - \tilde{W}_\ell^{(k)}\right)\tilde{X}_{\ell-1}\right\|_F \leq \left\|\tilde{W}_\ell - \tilde{W}_\ell^{(k)}\right\|_F (1 + \delta_\ell), \tag{13}$$

where $\tilde{W}_\ell = W_\ell (S_\ell^{\mathrm{stat}})^{-1}$ is the statically whitened weight matrix.

**Distribution-Aware Whitening.** Our proposed *Distribution-Aware Whitening* dynamically computes $S_\ell$ from the actual compressed inputs $\hat{X}_{\ell-1}$, thereby eliminating whitening drift ($\delta_\ell \approx 0$) and reducing anisotropy-induced distortions. Under this scheme, the compression error simplifies to:

$$E_\ell^{\mathrm{dist}} = \left\|\left(\tilde{W}_\ell - \tilde{W}_\ell^{(k)}\right)\tilde{X}_{\ell-1}\right\|_F = \left\|\tilde{W}_\ell - \tilde{W}_\ell^{(k)}\right\|_F, \tag{14}$$

where both the whitening matrix and the weight approximation are computed with respect to the current input distribution at each layer.

**Global Error Accumulation.** Assuming the activation function at each layer has Lipschitz constant $\rho_\ell$, the cumulative reconstruction error up to depth $L$ can be bounded by: $E^{\mathrm{total}} \leq \sum_{\ell=1}^L \left(\prod_{j=\ell+1}^L \rho_j\right)\rho_\ell E_\ell$. Substituting the error expressions under different whitening schemes in Eq. (9), Eq. (13) and Eq. (14) gives:

$$E_{\mathrm{raw}}^{\mathrm{total}} \leq \sum_{\ell=1}^L \left(\prod_{j=\ell+1}^L \rho_j\right)\rho_\ell \left\|(W_\ell - \hat{W}_\ell^{\mathrm{raw}})X_{\ell-1}\right\|_F, \tag{15}$$

$$E_{\mathrm{stat}}^{\mathrm{total}} \leq \sum_{\ell=1}^L \left(\prod_{j=\ell+1}^L \rho_j\right)\rho_\ell \left\|\tilde{W}_\ell - \tilde{W}_\ell^{(k)}\right\|_F \cdot \sqrt{1 + \delta_\ell}, \tag{16}$$

$$E_{\mathrm{dist}}^{\mathrm{total}} \leq \sum_{\ell=1}^L \left(\prod_{j=\ell+1}^L \rho_j\right)\rho_\ell \left\|\tilde{W}_\ell - \tilde{W}_\ell^{(k)}\right\|_F. \tag{17}$$

**The Advantage of Distribution Awareness.** It follows that the error gap between static and distribution-aware whitening is:

$$\Delta_{\mathrm{total}} := E_{\mathrm{stat}}^{\mathrm{total}} - E_{\mathrm{dist}}^{\mathrm{total}} \leq \sum_{\ell=1}^L \left(\prod_{j=\ell+1}^L \rho_j\right)\rho_\ell \left\|\tilde{W}_\ell - \tilde{W}_\ell^{(k)}\right\|_F \left(\sqrt{1 + \delta_\ell} - 1\right). \tag{18}$$

This bound quantifies the cumulative harm of failing to adapt whitening to the evolving feature distribution. Even modest spectral deviations $\delta_\ell$ at early layers can propagate nonlinearly due to the multiplicative nature of the network dynamics. In contrast, our distribution-aware approach minimizes such distortions by continually re-aligning the compressed input statistics, ensuring that compression remains both locally accurate and globally stable. See Appendix.D for details.

Table 1: Perplexity comparison of different methods across different compression ratios.

| Model | Method | WikiText-2 | | | PTB | | | C4 | | |
|---|---|---|---|---|---|---|---|---|---|---|
| | | 0.3 | 0.4 | 0.5 | 0.3 | 0.4 | 0.5 | 0.3 | 0.4 | 0.5 |
| LLaMA-7B | ASVD | 95.268 | 9111.411 | 37479.324 | 200.937 | 19425.612 | 57294.849 | 86.269 | 8676.642 | 37767.01 |
| | FWSVD | 33.001 | 199.142 | 4622.404 | 53.587 | 332.344 | 8861.445 | 38.24 | 255.026 | 9240.634 |
| | SVD-LLM | 9.526 | 13.854 | 26.864 | 28.967 | 63.864 | 191.380 | 26.390 | 57.281 | 153.840 |
| | Ours | **9.402** | **13.113** | **22.396** | **24.770** | **54.760** | **135.160** | **25.591** | **48.662** | **109.341** |
| Vicuna-7B | ASVD | 91.388 | 1580.427 | 22934.960 | 415.615 | 3069.448 | 28252.915 | 136.157 | 1735.991 | 24201.540 |
| | FWSVD | 43.690 | 347.362 | 4449.084 | 239.318 | 1711.730 | 9353.528 | 64.753 | 461.874 | 4421.256 |
| | SVD-LLM | 12.416 | 18.346 | 35.569 | 124.506 | 261.100 | 615.591 | 39.528 | 77.706 | 185.780 |
| | Ours | **11.936** | **16.572** | **26.511** | **95.419** | **182.380** | **352.582** | **34.162** | **61.070** | **118.667** |
| DeepSeek-7B | ASVD | 85.169 | 3806.825 | 64971.820 | 87.709 | 7580.528 | 99927.992 | 79.853 | 4355.394 | 57731.498 |
| | FWSVD | 68.416 | 202.822 | 600.397 | 99.775 | 265.391 | 890.699 | 118.319 | 325.196 | 774.628 |
| | SVD-LLM | 10.841 | 14.449 | 22.660 | 30.747 | 55.803 | 132.187 | 32.622 | 58.199 | 117.580 |
| | Ours | **10.809** | **14.069** | **20.499** | **29.273** | **49.925** | **96.091** | **31.214** | **52.398** | **94.887** |
| LLaMA-13B | ASVD | 17.648 | 201.027 | 11445.274 | 32.963 | 286.850 | 13304.711 | 20.866 | 183.898 | 10897.571 |
| | FWSVD | 12.963 | 45.150 | 193.531 | 22.123 | 75.662 | 275.487 | 18.509 | 64.610 | 245.409 |
| | SVD-LLM | **7.618** | 17.823 | 18.825 | **9.836** | 34.222 | 33.328 | **14.982** | 89.288 | 68.417 |
| | Ours | 7.635 | **17.393** | **18.653** | 9.930 | **29.628** | **32.252** | 15.099 | **65.343** | **65.838** |
| Vicuna-13B | ASVD | 28.309 | 189.392 | 1220.318 | 637.196 | 1601.441 | 3922.871 | 39.799 | 190.550 | 1080.129 |
| | FWSVD | 32.715 | 106.356 | 304.176 | 310.304 | 668.389 | 1475.578 | 47.408 | 146.253 | 395.776 |
| | SVD-LLM | 9.616 | 13.221 | 23.607 | 145.715 | 324.892 | 699.598 | 29.204 | 58.335 | 140.584 |
| | Ours | **9.349** | **12.447** | **19.244** | **132.653** | **230.471** | **430.424** | **27.809** | **51.640** | **111.015** |
| Qwen2-1.5B | SVD-LLM | 43.597 | 108.584 | 311.243 | 358.038 | 1044.427 | 3318.722 | 281.344 | 688.116 | 1794.847 |
| | Ours | **35.076** | **63.050** | **114.834** | **164.126** | **319.454** | **651.513** | **197.564** | **375.296** | **621.605** |
| Qwen2-7B | SVD-LLM | 14.817 | 28.154 | 54.368 | 119.290 | 243.933 | 591.097 | 83.436 | 161.216 | 336.788 |
| | Ours | **13.038** | **19.051** | **28.725** | **86.220** | **137.506** | **210.971** | **63.686** | **107.805** | **179.230** |
| Qwen2.5-7B | SVDLLM | 17.889 | 31.461 | 68.496 | 156.168 | 356.669 | 987.733 | 85.303 | 187.262 | 454.095 |
| | Ours | **15.240** | **23.594** | **36.115** | **130.661** | **224.162** | **368.287** | **71.549** | **129.353** | **245.203** |
| Qwen2.5-14B | SVDLLM | 11.876 | 20.306 | 62.108 | 92.846 | 279.756 | 919.945 | 56.899 | 147.133 | 384.962 |
| | Ours | **11.016** | **14.842** | **23.124** | **69.025** | **121.101** | **210.055** | **44.998** | **80.006** | **149.412** |
| Qwen2.5-32B | SVDLLM | 30.962 | 38.669 | 46.788 | 681.311 | 780.162 | 1196.067 | 344.032 | 381.856 | 483.195 |
| | Ours | **19.914** | **22.785** | **30.869** | **258.474** | **302.2099** | **413.135** | **224.699** | **286.357** | **379.361** |
| LLaMA3-8B | SVD-LLM | 33.176 | 154.899 | 3908.397 | 502.888 | 4378.216 | 13925.224 | 361.046 | 1558.488 | 5595.825 |
| | Ours | **23.702** | **43.850** | **83.950** | **104.786** | **239.840** | **599.995** | **232.641** | **451.045** | **687.607** |

## 4 EXPERIMENTS

The goal of our method is to preserve task-relevant information under a fixed parameter budget, thereby improving compression fidelity. To comprehensively assess the effectiveness of our distribution-aware low-rank approximation framework, we organize our experiments as follows. **Subsection 4.1** describes the experimental setup. **Subsection 4.2** compares our method with existing SVD-based compression approaches under fixed compression ratios, showing consistent gains in performance preservation. **Subsection 4.2.1** investigates the compatibility with quantization, **Subsection 4.2.2** investigates the effect of dynamic rank allocation, **Subsection 4.2.3** investigates the compatibility with post-compression fine-tuning, and **Subsection 4.2.4** provides a comparison with existing pruning methods. **Additional discussions**—covering the impact of calibration set selection, the zero-shot accuracy of compressed models, the extension to large vision-language models and the analysis of inference overhead—are provided in the Appendix. **Subsection 4.3** further evaluates our method as an initialization strategy for LoRA-style fine-tuning, demonstrating consistent outperformance over both conventional Kaiming initialization (Hu et al., 2022) and the recently proposed PiSSA (Meng et al., 2024).

### 4.1 EXPERIMENTAL SETUP

**Foundation LLMs.** We conducted experiments on existing popular LLMs at various scales, including LLaMA-{7B, 13B} (Touvron et al., 2023), Vicuna-v1.5-{7B, 13B} (Chiang et al., 2023), Deepseek-7B (DeepSeek-AI et al., 2025), Qwen2-{3B, 7B} (Yang et al., 2024), LLaMA3-8B (Dubey et al., 2024), and Qwen2.5-{7B, 14B, 32B} (Qwen et al., 2025).

**Baselines.** For post-training compression, we compare our method against several representative SVD-based techniques, including FWSVD (Hsu et al., 2022), ASVD (Yuan et al., 2023), and SVD-LLM (Wang et al., 2025b). For parameter-efficient fine-tuning, we consider LoRA (Hu et al., 2022)

with standard Kaiming initialization and the recently proposed PiSSA initialization (Meng et al., 2024) as strong baselines.

**Evaluation Setup.** To evaluate language modeling performance, we report perplexity on several standard language modeling benchmarks, including WikiText-2 (Merity et al., 2016), Penn Treebank (PTB) (Marcus et al., 1993), and C4 (Mihaylov et al., 2018). We assess zero-shot accuracy on the MMLU (Hendrycks et al., 2021) Benchmark to further demonstrate the effectiveness of our proposed method in Appendix.J. In the parameter-efficient fine-tuning setting, we consider three task domains: Mathematical reasoning: Models are fine-tuned on 1000 samples from MetaMathQA dataset (Yu et al., 2023) and evaluated on the GSM8K (Cobbe et al., 2021) and MATH (Hendrycks et al., 2021) validation sets; Code generation: Models are fine-tuned on 1000 samples from Code Feedback dataset and evaluated on HumanEval (Chen et al., 2021) and HumanEval+ (Liu et al., 2023); Abstract concept understanding: Models are trained on 500 samples from Object Counting dataset (Srivastava et al., 2023) and evaluated on its validation set to assess their ability to handle abstract visual-numeric concepts. Unless otherwise noted, all experimental results are obtained by directly applying compression without any post-compression fine-tuning. In the main experimental settings, the same compression ratio is applied to every module of the model, except in Section 4.2.2, where dynamic compression ratio allocation is considered. All experiments are conducted on 4 NVIDIA A100 GPUs, each with 40 GB of memory. Code will be released to facilitate reproducibility and further research.

## 4.2 COMPRESSION PERFORMANCE COMPARISON

Perplexity measures a model's uncertainty in predicting the next token and a lower perplexity indicates greater confidence and stronger language modeling capabilities. We adopt perplexity (PPL) as the primary metric for evaluating post-compression language modeling quality, following standard practice in model compression literature (Wang et al., 2025b). Table 1 compares the performance of various SVD-based compression methods across eight large language models (LLMs) of different sizes and architectures, evaluated on standard language modeling benchmarks including WikiText-2, PTB, and C4. Notably, methods such as ASVD and FWSVD exhibit limited compatibility and suboptimal performance when extended to recent LLMs. Therefore, for newer models like LLaMA3, we only compare against SVD-LLM, the current state-of-the-art SVD-based method. Our method consistently outperforms existing baselines in most scenarios, demonstrating the effectiveness of distribution-awareness in mitigating compression-induced degradation. For example, under a 30% compression ratio on the WikiText-2 task, our method achieves a 9.464 lower perplexity than SVD-LLM on the LLaMA3-8B model. Additionally, we report the perplexity comparison under a mild compression ratio of 0.2 in Appendix F, and the zero-shot accuracy on the MMLU benchmark in Appendix J, to further validate the advantages of our proposed method.

### 4.2.1 COMPATIBILITY WITH QUANTIZATION

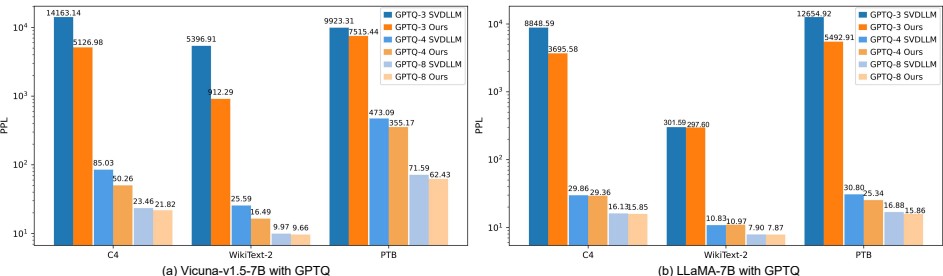

Figure 1: Comparison of perplexity after post-training quantization (GPTQ-8bit, 4bit, and 3bit) applied to models compressed by our method and SVD-LLM. Our method consistently shows less degradation, particularly under more aggressive quantization.

Beyond parameter compression, quantization remains a key technique for model size and inference efficiency reduction. In this subsection, we evaluate whether our distribution-aware low-rank approximation method is compatible with post-compression quantization. Specifically, we compare

our method with the strongest SVD-based baseline, SVD-LLM at a compression ratio of 20%, under various quantization settings using the GPTQ (Frantar et al., 2023) framework. As shown in Figure 1, under milder quantization settings (e.g., GPTQ-8Bit), both methods are able to maintain performance well. On the C4 dataset with the Vicuna-V1.5-7B model, the perplexity of SVD-LLM is only 0.31 higher than that of our method. However, under more aggressive quantization (e.g., GPTQ-4Bit), our method demonstrates substantially stronger robustness. For instance, with GPTQ-4Bit, the perplexity of SVD-LLM increases sharply to 85.03, which is 34.77 worse than that of our method. These results indicate that our approach is more compatible with quantization, particularly under aggressive compression regimes. This highlights an additional advantage of aligning input distributions prior to low-rank approximation: better numerical stability and robustness under further quantization of the compressed model.

### 4.2.2 COMPATIBILITY WITH DYNAMIC COMPRESSION RATIOS ALLOCATION

Beyond designing better low-rank approximation algorithms to retain critical information under parameter constraints, another line of work (Ding et al., 2025; Wang et al., 2025a) focuses on improving compression quality by assigning different compression ratios to different layers based on their relative importance. Our method, which reduces the mismatch between the compressed and original model from a distributional perspective, is inherently compatible with such strategies. Intuitively, better-informed compression ratio allocation should complement our method and lead to further performance gains. To verify this, we follow the dynamic compression ratio configuration recommended by SVM-LLM-V2(Wang et al., 2025a) and evaluate its effectiveness in combination with our method. As shown in Table 2, introducing layerwise compression ratio allocation leads to consistent improvements for both our approach and the strongest existing baseline, SVD-LLM (Wang et al., 2025b). Across different global compression ratios, our method consistently outperforms SVD-LLM, demonstrating its compatibility with advanced allocation strategies.

Table 2: Comparison of our method and SVD-LLM under dynamic compression ratio allocation. Results show that both methods benefit from dynamic allocation, while our approach consistently achieves better performance across different global compression ratios.

| Compression Ratio | Method | Vicuna-V1.5-7B | | | LLaMA-7B | | |
|---|---|---|---|---|---|---|---|
| | | WikiText-2 | PTB | C4 | WikiText-2 | PTB | C4 |
| 0.2 | SVD-LLM | 10.0075 | 65.0998 | 21.4812 | 8.0148 | 17.1674 | 15.3974 |
| | Ours | **9.7239** | **59.2590** | **20.2759** | **7.8867** | **16.2424** | **15.1916** |
| 0.3 | SVD-LLM | 12.1873 | 94.8598 | 31.7581 | 9.5549 | 26.5252 | 22.4992 |
| | Ours | **11.5799** | **80.6119** | **29.2213** | **9.2750** | **24.9149** | **21.6804** |
| 0.4 | SVD-LLM | 17.0807 | 184.5728 | 56.0702 | 13.0047 | 54.2788 | 39.5939 |
| | Ours | **15.2986** | **132.9584** | **47.6044** | **12.1296** | **46.5087** | **35.6820** |
| 0.5 | SVD-LLM | 28.8035 | 512.3306 | 137.0720 | 21.8262 | 142.8860 | 111.9017 |
| | Ours | **22.2192** | **316.9484** | **92.7974** | **18.7572** | **102.1029** | **84.5901** |

### 4.2.3 COMPATIBILITY WITH LIGHTWEIGHT POST-COMPRESSION FINE-TUNING

Lightweight Post-Compression Fine-Tuning is a widely used approach (Ma et al., 2023; Wang et al., 2025b) to recover performance after model compression. While the previous section discussed model performance immediately after compression, Table 3 presents a comparison of the post-compression fine-tuning results of our method and SVD-LLM at a compression ratio of 0.5. For a fair comparison, we adopt the same experimental settings as those used in SVD-LLM. *Partial Update* refers to updating only $W_{u,\ell}$ in Eq. (8), whereas *Full Update* updates $W_{u,\ell}$ and subsequently $W_{v,\ell}$. Our method outperforms the baseline SVD-LLM after lightweight post-compression fine-tuning across various language models in most cases. For example, with full update on the LLaMA3-8B model, the perplexity of our method on the PTB dataset is 64.5531, which is 63.6923 points lower than that of SVD-LLM (128.2454). See additional results for other compression ratios in Appendix.K.

Table 3: Perplexity of our method across three language models under different compression ratios. After lightweight post-compression fine-tuning, our proposed method outperforms the baseline method SVD-LLM in most cases.

| Method | Vicuna-7B | | | Qwen2-7B | | | LLaMA3-8B | | |
|---|---|---|---|---|---|---|---|---|---|
| | WikiText-2 | PTB | C4 | WikiText-2 | PTB | C4 | WikiText-2 | PTB | C4 |
| SVD-LLM Partial Update | 14.7036 | 130.6588 | 26.8014 | 22.4540 | 74.3111 | 54.7868 | 51.7419 | 193.0543 | 124.1097 |
| Ours Partial Update | **14.3285** | **108.6498** | **25.6585** | **20.2631** | **58.0048** | **46.4589** | **36.6911** | **75.2853** | **73.5259** |
| SVD-LLM Full Update | 12.7389 | 180.4358 | 21.5852 | 16.3075 | 46.8237 | 39.3111 | 49.4664 | 128.2454 | 85.4848 |
| Ours Full Update | **12.6762** | **124.4990** | **21.5079** | **16.0868** | **40.7479** | **35.7692** | **32.1073** | **64.5531** | **64.6075** |

### 4.2.4 COMPARISON WITH PRUNING-BASED APPROACHES

Model pruning is another pathway for model compression, aiming to reduce computational and memory overhead by removing redundant weights or structures. This naturally raises the question of whether our SVD-based compression approach can be comparable to pruning-based methods. In Figure 2, we compare our method with two representative pruning approaches: LLM-Pruner (Ma et al., 2023), which restores performance through lightweight post-compression fine-tuning, and FLAP (An et al., 2024), which applies a baseline bias compensation strategy after pruning. For a fair comparison, we also apply lightweight post-compression fine-tuning to our method. The horizontal dashed line in the figure indicates the performance of the uncompressed original model. We observe that across different compression ratios, our method consistently achieves better language modeling performance (i.e., lower perplexity) than both FLAP and LLM-Pruner on the WikiText-2 task. Interestingly, on the PTB and C4 benchmarks, our method performs worse than pruning-based approaches at lower compression ratios, but surpasses them at higher compression levels. This suggests that our method may be particularly effective in more aggressive compression scenarios.

It is important to note that pruning, quantization, and low-rank approximation represent distinct pathways for model compression, each with its own trade-offs and strengths. Our intention is not to claim absolute superiority over other compression types, but rather to demonstrate that the proposed distribution-aware low-rank approximation framework is a competitive and promising alternative.

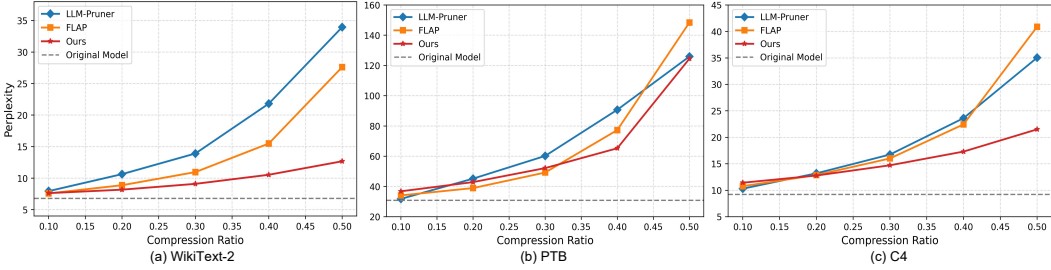

Figure 2: Comparison of perplexity across different compression ratios using three model compression strategies on Vicuna-V1.5-7B: our distribution-aware SVD-based method, FLAP, and LLM-Pruner.

### 4.3 LoRA INITIALIZATION COMPARISON

This subsection compares three LoRA initialization strategies: the vanilla Kaiming initialization, the PiSSA initialization, and our proposed distribution-aware initialization method on natural language generation tasks. Unlike PiSSA, which directly uses the principal components from the SVD decomposition as initialization, our method takes the input data distribution into account to more accurately determine which components in the weight matrix are most informative, and uses these as initialization. As shown in Table 4, our method consistently achieves better performance across mathematical reasoning, code generation, and object counting tasks. For example, on the GSM8K task with a rank of 256, our method outperforms PiSSA by 6.4% on the Vicuna-V1.5-7B model. An interesting question is whether such fine-tuning leads to significant forgetting on other unrelated tasks. We explore this in Appendix N through additional experiments on the MMLU benchmark.

Table 4: Performance comparison of vanilla LoRA, PiSSA, and our initialization method.

| Model | Rank | Method | GSM8K | Math | HumanEval | HumanEval+ | Obj. Count | Avg. |
|-------|------|--------|-------|------|-----------|------------|------------|------|
| Vicuna-V1.5-7B | 256 | Vanilla | 0.293 | 0.034 | 0.177 | 0.146 | 0.324 | 0.1948 |
| | | PiSSA | 0.320 | 0.042 | 0.189 | 0.171 | 0.576 | 0.2596 |
| | | Ours | **0.384** | **0.046** | **0.213** | **0.183** | **0.644** | **0.2940** |
| | 128 | Vanilla | 0.306 | 0.036 | 0.159 | 0.128 | 0.328 | 0.1914 |
| | | PiSSA | 0.331 | 0.043 | 0.207 | 0.183 | 0.572 | 0.2672 |
| | | Ours | **0.384** | **0.049** | **0.220** | **0.195** | **0.620** | **0.2936** |
| | 64 | Vanilla | 0.301 | 0.034 | 0.165 | 0.134 | 0.324 | 0.1916 |
| | | PiSSA | 0.319 | 0.042 | 0.183 | 0.165 | 0.612 | 0.2642 |
| | | Ours | **0.371** | **0.046** | **0.183** | **0.165** | **0.616** | **0.2762** |
| Qwen2-7B | 256 | Vanilla | 0.801 | 0.458 | 0.622 | 0.579 | 0.444 | 0.5808 |
| | | PiSSA | 0.805 | **0.462** | 0.726 | 0.683 | 0.680 | 0.6712 |
| | | Ours | **0.818** | 0.460 | **0.726** | **0.683** | **0.848** | **0.7070** |
| | 128 | Vanilla | 0.803 | 0.459 | 0.652 | 0.598 | 0.476 | 0.5976 |
| | | PiSSA | 0.810 | 0.461 | 0.659 | 0.622 | 0.780 | 0.6664 |
| | | Ours | **0.810** | **0.461** | **0.689** | **0.646** | **0.840** | **0.6892** |
| | 64 | Vanilla | 0.810 | 0.458 | 0.622 | 0.573 | 0.524 | 0.5974 |
| | | PiSSA | 0.806 | 0.460 | 0.707 | 0.671 | 0.792 | 0.6872 |
| | | Ours | **0.821** | **0.465** | **0.726** | **0.671** | **0.824** | **0.7014** |

In Figure 3 we examine how the three LoRA initialization strategies affect training convergence on both mathematical (MetaMathQA) and programming (Code Feedback) tasks. Our initialization yields substantially lower initial loss than vanilla initialization and drops rapidly within the first 100 steps, ultimately converging to a slightly lower final loss. In contrast, standard LoRA requires a longer warm-up period to reach comparable loss levels. These observations indicate that our initialization not only accelerates convergence and further reduces training loss, but also achieves better final model performance. Additional visualizations and results with other ranks are provided in Appendix L.

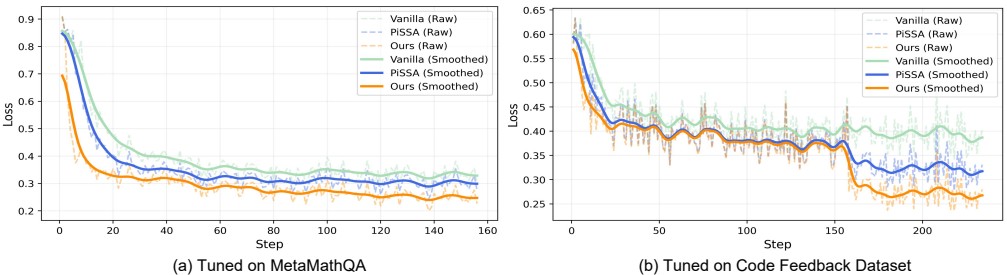

(a) Tuned on MetaMathQA

(b) Tuned on Code Feedback Dataset

Figure 3: Training loss for fine-tuning Vicuna-V1.5-7B with LoRA (rank 256) under three different initialization schemes: Vanilla Kaiming Initialization, PiSSA, and our proposed method. (a) Results on the MetaMathQA dataset; (b) results on the Code Feedback dataset. Dashed lines represent raw training loss values, while solid lines denote Gaussian-smoothed curves for improved visual clarity.

## 5 CONCLUSION

Low-rank approximation has long been a foundational technique for reducing the computational and memory demands of large models. However, conventional methods often rely on an implicit assumption of stable and isotropic input distributions, overlooking the distributional drift that naturally arises across layers during approximation. To address this, we propose a *distribution-aware whitening framework* that dynamically normalizes each layer's input based on its evolving activation statistics. By ensuring that low-rank projections operate in an approximately isotropic feature space, our method decouples approximation error from input covariance, thereby yielding more faithful and stable approximations. We provide both theoretical insights into the relationship between input misalignment and error amplification, as well as empirical validation across diverse models and tasks in both post-training compression and parameter-efficient fine-tuning. Beyond demonstrating consistent performance improvements, this work underscores the need for future research to explicitly consider input distribution characteristics when developing higher-fidelity approximation methods.

**Ethics Statement** This study adheres to the ICLR Code of Ethics. We exclusively use publicly available datasets (e.g., WikiText2, PTB, C4), none of which contain personally identifiable information. We have disclosed all relevant details and ensured research integrity in accordance with the Code of Ethics.

**Reproducibility Statement** We have taken multiple measures to ensure reproducibility. All experimental settings, including datasets, hyperparameters, model configurations, and evaluation metrics, are detailed in the main text and Appendix. Ablation studies are also included. These resources enable other researchers to reproduce our results and verify the conclusions presented in this paper. We will make our method publicly available as open source in the future to facilitate further research and provide valuable reference for the community.

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

## A  LIMITATIONS

This work focuses on leveraging input distribution statistics to improve the approximation fidelity of low-rank methods. The proposed framework can serve as a drop-in replacement for conventional SVD used in model compression, as well as in SVD-based initialization strategies such as PiSSA for LoRA. While our method shows improvements across diverse settings, we acknowledge that, in line with the *No Free Lunch Theorem*, no single approach can universally dominate across all tasks and deployment scenarios. We outline several limitations of our method: 1) Compared to standard SVD, our method introduces additional computational overhead due to the per-layer whitening operations; 2) The performance may be influenced by the choice and distribution of the calibration data. In our experiments, we adopt the 256-sample subset of WikiText-2 as suggested by ASVD. However, identifying optimal or adaptive calibration strategies remains a non-trivial and open research direction; 3) Although we evaluate our method across eight diverse models with varying sizes and architectures, we are currently unable to conduct experiments on very large-scale models (e.g., 65B or 70B parameters) due to computational constraints. Nonetheless, our method is architecture-agnostic and theoretically scalable, and we will release our code to support further evaluation and broader adoption by the community.

## B  THE USE OF LARGE LANGEUAGE MODELS

In this work, a large language model was employed solely for language refinement, aiming to produce more fluent expressions and to avoid grammatical issues. The model was not used for content generation. The rapid development of large models provides non-native English speakers with a valuable tool to better present their work. We hope that our research can make a modest contribution along the fast-evolving path of large model development.

## C  PROOF OF WHITENED COMPRESSION ERROR EQUIVALENCE

By definition, the whitening matrix

$$S_\ell = \left( X_{\ell-1} X_{\ell-1}^\top + \epsilon I \right)^{-\frac{1}{2}} \tag{19}$$

transforms the original inputs $X_{\ell-1} \in \mathbb{R}^{d_{\ell-1} \times n}$ into whitened inputs

$$\tilde{X}_{\ell-1} = S_\ell X_{\ell-1}, \tag{20}$$

which satisfy (approximately)

$$\tilde{X}_{\ell-1} \tilde{X}_{\ell-1}^\top \approx I, \tag{21}$$

where $n$ is the number of input samples and $I \in \mathbb{R}^{d_{\ell-1} \times d_{\ell-1}}$ is the identity matrix.

Correspondingly, the transformed weight matrix is defined as

$$\tilde{W}_\ell = W_\ell S_\ell^{-1}. \tag{22}$$

We perform low-rank compression on $\tilde{W}_\ell$ via truncated singular value decomposition (SVD), obtaining the best rank-$k$ approximation

$$\tilde{W}_\ell^{(k)} = \arg \min_{\text{rank}(\cdot) \leq k} \|\tilde{W}_\ell - \cdot\|_F. \tag{23}$$

The compressed weight in the original space is then recovered as

$$\hat{W}_\ell = \tilde{W}_\ell^{(k)} S_\ell. \tag{24}$$

To analyze the output reconstruction error, consider

$$\|W_\ell X_{\ell-1} - \hat{W}_\ell X_{\ell-1}\|_F = \|W_\ell X_{\ell-1} - \tilde{W}_\ell^{(k)} S_\ell X_{\ell-1}\|_F. \tag{25}$$

Substituting $\tilde{X}_{\ell-1} = S_\ell X_{\ell-1}$ and $W_\ell = \tilde{W}_\ell S_\ell$, we rewrite this as

$$\|\tilde{W}_\ell \tilde{X}_{\ell-1} - \tilde{W}_\ell^{(k)} \tilde{X}_{\ell-1}\|_F = \|(\tilde{W}_\ell - \tilde{W}_\ell^{(k)})\tilde{X}_{\ell-1}\|_F. \tag{26}$$

Using the Frobenius norm property and the whitened input covariance, we have

$$\|(\tilde{W}_\ell - \tilde{W}_\ell^{(k)})\tilde{X}_{\ell-1}\|_F^2 = \text{trace}\left(\tilde{X}_{\ell-1}^\top (\tilde{W}_\ell - \tilde{W}_\ell^{(k)})^\top (\tilde{W}_\ell - \tilde{W}_\ell^{(k)})\tilde{X}_{\ell-1}\right). \tag{27}$$

Since $\tilde{X}_{\ell-1}\tilde{X}_{\ell-1}^\top = I$, the above reduces to

$$\text{trace}\left((\tilde{W}_\ell - \tilde{W}_\ell^{(k)})(\tilde{W}_\ell - \tilde{W}_\ell^{(k)})^\top\right) = \|\tilde{W}_\ell - \tilde{W}_\ell^{(k)}\|_F^2. \tag{28}$$

By the Eckart-Young theorem, the truncated $\tilde{W}_\ell^{(k)}$ yields the optimal rank-$k$ approximation, and the approximation error in Frobenius norm is given by the square root of the sum of the squares of the discarded singular values:

$$\|\tilde{W}_\ell - \tilde{W}_\ell^{(k)}\|_F = \left(\sum_{i=k+1}^{r} \sigma_i^2\right)^{1/2}, \tag{29}$$

where $\sigma_i$ are the singular values of $\tilde{W}_\ell$ sorted in descending order, and $r = \min(d_\ell, d_{\ell-1})$.

Therefore, performing low-rank approximation in the whitened feature space guarantees that the output reconstruction error directly corresponds to the truncated SVD error of the whitened weight matrix, which explicitly takes the input covariance structure into account.

# D ERROR PROPAGATION UNDER LAYERWISE COMPRESSION

This section formally derives the total reconstruction error of a deep model under low-rank layerwise approximation.

## D.1 NOTATION

Consider a model with $L$ layers. At each layer $\ell$, the pre- and post-activation outputs can be fomulated as:

$$Z_\ell = W_\ell X_{\ell-1}, \qquad X_\ell = \phi_\ell(Z_\ell), \tag{30}$$

where $\phi_\ell$ is $\rho_\ell$-Lipschitz:

$$\|\phi_\ell(a) - \phi_\ell(b)\|_F \le \rho_\ell \|a - b\|_F, \quad \forall a, b. \tag{31}$$

Assume each weight matrix $W_\ell$ is compressed to a rank-$k_\ell$ approximation $\hat{W}_\ell$, producing compressed outputs $\hat{X}_\ell$ through a forward pass.

Define the final-layer reconstruction error:

$$E^{\text{total}} := \|X_L - \hat{X}_L\|_F. \tag{32}$$

## D.2 RECURSIVE ERROR PROPAGATION

We analyze layerwise errors recursively. Using the Lipschitz property of $\phi_\ell$ and the triangle inequality in Eq. (31):

$$\|X_\ell - \hat{X}_\ell\|_F = \|\phi_\ell(W_\ell X_{\ell-1}) - \phi_\ell(\hat{W}_\ell \hat{X}_{\ell-1})\|_F \le \rho_\ell \|W_\ell X_{\ell-1} - \hat{W}_\ell \hat{X}_{\ell-1}\|_F. \tag{33}$$

Decompose the error into:

$$\|W_\ell X_{\ell-1} - \hat{W}_\ell \hat{X}_{\ell-1}\|_F \le \underbrace{\|W_\ell X_{\ell-1} - \hat{W}_\ell X_{\ell-1}\|_F}_{E_\ell:\text{compression error}} + \underbrace{\|\hat{W}_\ell(X_{\ell-1} - \hat{X}_{\ell-1})\|_F}_{\text{propagated error}}. \tag{34}$$

Let $B_\ell := \|\hat{W}_\ell\|_2$. Then:

$$\|X_\ell - \hat{X}_\ell\|_F \leq \rho_\ell \left( E_\ell + B_\ell \|X_{\ell-1} - \hat{X}_{\ell-1}\|_F \right). \tag{35}$$

Unfolding this recursion yields a bound on the total output error:

$$E^{\text{total}} \leq \sum_{\ell=1}^{L} \left( \prod_{j=\ell+1}^{L} \rho_j B_j \right) \rho_\ell E_\ell. \tag{36}$$

In the case of spectral normalization or bounded operator norm ($B_j \leq 1$), this simplifies to:

$$E^{\text{total}} \leq \sum_{\ell=1}^{L} \left( \prod_{j=\ell+1}^{L} \rho_j \right) \rho_\ell E_\ell. \tag{37}$$

### D.3 INFLUENCE OF DIFFERENT INPUT WHITENING STRATEGIES

This subsection analyzes the influence of different input whitening strategies on the per-layer error terms $E_\ell$:

**(a) Raw compression (no whitening):**

$$E_\ell^{\text{raw}} = \|W_\ell X_{\ell-1} - \hat{W}_\ell^{\text{raw}} X_{\ell-1}\|_F.$$

**(b) Static whitening:** A fixed whitening matrix $S_\ell^{\text{stat}}$ is computed from uncompressed features. Define the whitened input:

$$\tilde{X}_{\ell-1} = S_\ell^{\text{stat}} \hat{X}_{\ell-1}, \quad \tilde{X}_{\ell-1}\tilde{X}_{\ell-1}^\top = I + \Delta_\ell, \quad \delta_\ell := \|\Delta_\ell\|_2.$$

Then, using sub-multiplicativity of norms, we have:

$$E_\ell^{\text{stat}} = \|(\tilde{W}_\ell - \tilde{W}_\ell^{(k)})\tilde{X}_{\ell-1}\|_F \leq \|\tilde{W}_\ell - \tilde{W}_\ell^{(k)}\|_F \cdot \|\tilde{X}_{\ell-1}\|_2.$$

By definition of the whitened input, $\tilde{X}_{\ell-1}\tilde{X}_{\ell-1}^\top = I + \Delta_\ell$, and hence

$$\|\tilde{X}_{\ell-1}\|_2^2 = \|\tilde{X}_{\ell-1}\tilde{X}_{\ell-1}^\top\|_2 = \|I + \Delta_\ell\|_2 = 1 + \delta_\ell,$$

which gives

$$\|\tilde{X}_{\ell-1}\|_2 \leq \sqrt{1 + \delta_\ell}.$$

Substituting into the earlier bound, we obtain:

$$E_\ell^{\text{stat}} \leq \|\tilde{W}_\ell - \tilde{W}_\ell^{(k)}\|_F \cdot \sqrt{1 + \delta_\ell}.$$

**(c) Distribution-aware whitening (proposed):** Whitening is recomputed on the compressed input:

$$S_\ell = (\hat{X}_{\ell-1}\hat{X}_{\ell-1}^\top + \epsilon I)^{-1/2}, \quad \text{so that } \tilde{X}_{\ell-1}\tilde{X}_{\ell-1}^\top \approx I.$$

This yields a tight bound:

$$E_\ell^{\text{dist}} \approx \|\tilde{W}_\ell - \tilde{W}_\ell^{(k)}\|_F.$$

Plugging the respective per-layer errors into the total error bound:

$$E_{\text{raw}}^{\text{total}} \leq \sum_{\ell=1}^{L} \left( \prod_{j=\ell+1}^{L} \rho_j B_j \right) \rho_\ell E_\ell^{\text{raw}},$$

$$E_{\text{stat}}^{\text{total}} \leq \sum_{\ell=1}^{L} \left( \prod_{j=\ell+1}^{L} \rho_j B_j \right) \rho_\ell \|\tilde{W}_\ell - \tilde{W}_\ell^{(k)}\|_F \cdot \sqrt{1 + \delta_\ell},$$

$$E_{\text{dist}}^{\text{total}} \leq \sum_{\ell=1}^{L} \left( \prod_{j=\ell+1}^{L} \rho_j B_j \right) \rho_\ell \|\tilde{W}_\ell - \tilde{W}_\ell^{(k)}\|_F.$$

Distribution-aware whitening aligns the compression with the actual post-compression distribution, ensuring $\tilde{X}_{\ell-1}\tilde{X}_{\ell-1}^\top \approx I$ and effectively minimizing the inflation term $\sqrt{1 + \delta_\ell}$. This leads to more robust error propagation across layers.

# E    LAYER-WISE LIPSCHITZ ANALYSIS

In this section, we separately measure the Lipschitz constants for the query, key, value, and output projections in the attention sublayers, as well as for the two linear layers in the feed-forward sublayers. The measured constants across layers are visualized in Figure 4. Empirically, we observe the following patterns:

1) In attention sublayers, the query and key projections (Q/K) generally have higher Lipschitz constants than the value projections (V), reflecting their greater sensitivity to input perturbations.

2) In feed-forward sublayers, the first linear layer (W1) tends to have larger Lipschitz constants than the second (W2).

3) In deeper layers, particularly near the output, Lipschitz constants tend to increase slightly, implying that errors can be progressively amplified as they propagate toward the output.

These Lipschitz constants indicate that small errors can be amplified through the network, potentially harming model fidelity. Our distribution-aware whitening mechanism helps suppress such error propagation during low-rank compression.

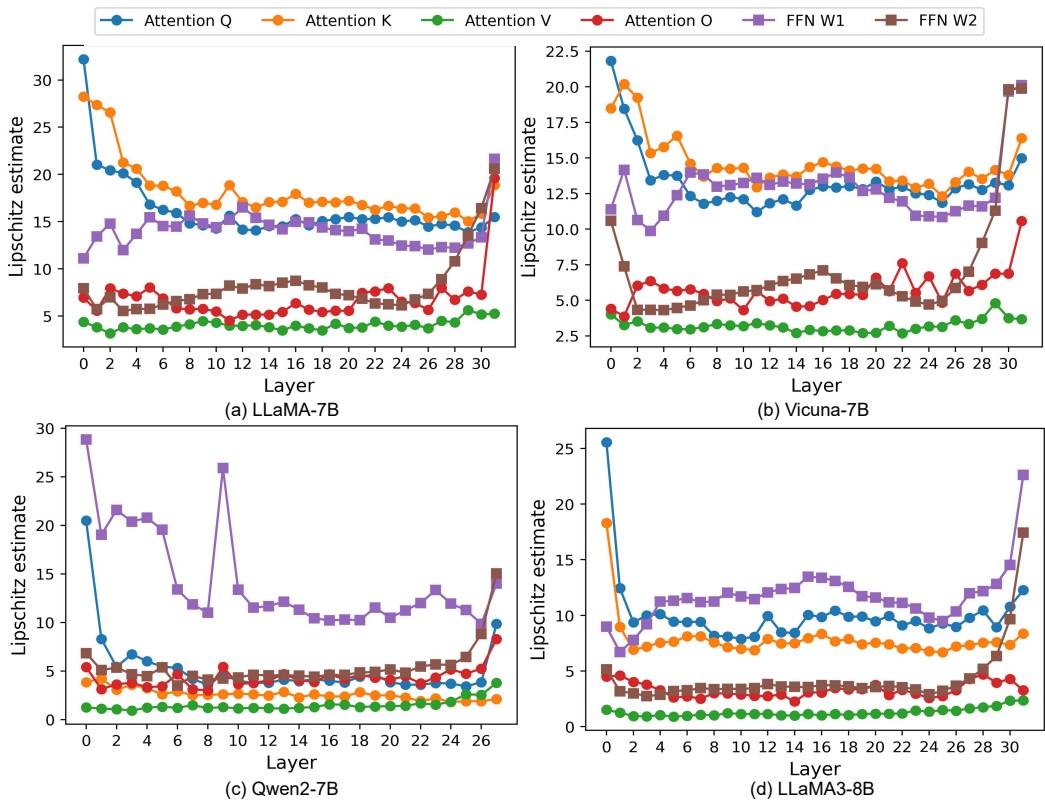

Figure 4: Layer-wise Lipschitz constants for different modules (attention and feed-forward) across various LLMs.

# F    PERFORMANCE COMPARISON UNDER A MILD COMPRESSION RATIO

Due to page limitations, the main text only reports the performance of different compression methods at compression ratios of 30%, 40%, and 50%. Here, we provide a comparison at a more moderate compression ratio of 20%. As shown in Figure 5, on the WikiText-2 task, our proposed method outperforms SVD-LLM across all models, although the margin is relatively small. By comparison, on PTB and C4, our method exhibits much clearer advantages. For example, when compressing the

Vicuna-7B model by 20%, our method achieves a PPL of 61.51 on PTB, which is 9.68 points better than SVD-LLM.

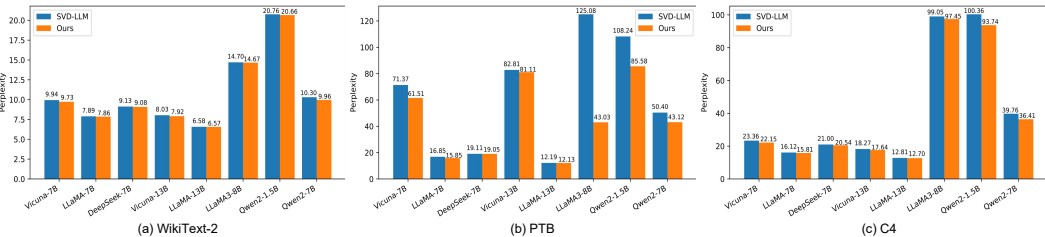

(a) WikiText-2      (b) PTB      (c) C4

Figure 5: Comparison between our method and SVD-LLM on different models when compressing 20% of parameters.

# G  ANALYSIS OF INFERENCE OVERHEAD

The computational complexity of a model primarily depends on its architecture, the number of parameters, input sequence length, and hardware implementation. In this section, we use the Vicuna-7B model as an example to analyze the computational savings introduced by our method, with experiments conducted in FP32 precision.

Our method achieves significant computational reduction by applying structured low-rank approximations to the weight matrices. For a given weight matrix $W \in \mathbb{R}^{m \times n}$, we perform a truncated SVD decomposition: $W \approx U_k \Sigma_k V_k^T$, where $k < \min(m, n)$ is the target rank determined by the compression ratio. This decomposition transforms the original matrix multiplication $WX$ (with $X \in \mathbb{R}^{n \times p}$ and $p$ denoting the number of input vectors) into two sequential operations:

1) $\hat{X} = \Sigma_k V_k^T X$ with approximately $knp$ FLOPs, and

2) $U_k \hat{X}$ with approximately $mkp$ FLOPs,

resulting in a total computational cost of $C_{UV} \approx k \cdot p \cdot (m + n)$, where we simplify by ignoring the factor of 2 from multiply–add operations and other small contributions. The original multiplication has cost $C_{WX} \approx mnp$ under the same simplification.

To select $k$ based on a desired parameter retention ratio $\alpha$ (e.g., $\alpha = 0.6$ for 60% of original parameters), we set: $k = \left\lfloor \frac{\alpha mn}{m+n} \right\rfloor$, ensuring $k < \min(m, n)$. The theoretical FLOPs reduction ratio is then $r = 1 - \frac{C_{UV}}{C_{WX}}$, ignoring additional computations from residual connections, LayerNorm, biases, and activation functions, which typically contribute a small fraction of total FLOPs.

Experiments in FP32 precision across different batch sizes (32, 64, 128), sequence lengths (32, 64, 128), and compression ratios (0.0–0.5) validate these savings (Fig. 6). As the compression ratio increase, memory usage drops (e.g., from 27.56GB to 15.49GB for sequence length 32, batch size 32, when increase ratio from 0.0 to 0.5), and throughput improves (from 475.19 to 598.59 tokens/sec). Larger batch sizes and longer sequences increase the absolute computational savings and throughput, as more input vectors benefit from the reduced computation.

# H  EXTENSION TO LARGE VISION-LANGUAGE MODELS

A natural question arises: since the proposed method is effective for various large language models, can this compression approach be extended to multimodal scenarios, such as large vision-language models (VLMs)? To investigate this, we conducted experiments on three VLMs: LLaVA-Next 7B, LLaVA-Next 13B, and LLaVA-V1.5 13B, using the ScienceQA task to evaluate the performance of the compressed models.

As shown in Table 5, compared to the standard SVD method, our approach introduces a whitening operation that effectively preserves important information during compression. This results in sig-

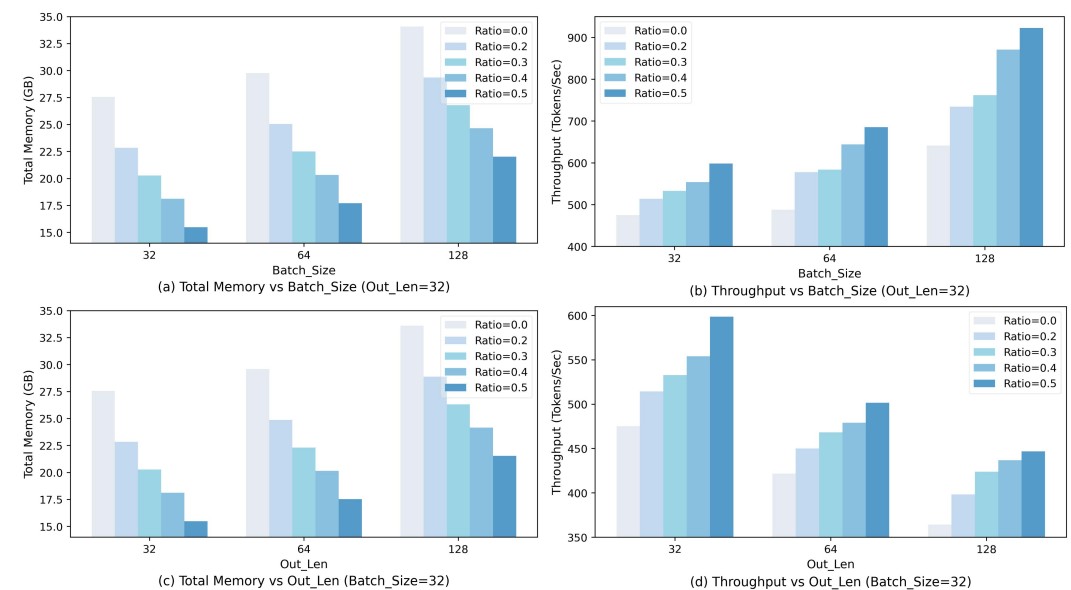

Figure 6: Memory usage (GB) and throughput (tokens/sec) of Vicuna-7B in FP32 precision under different compression ratios.

nificantly smaller performance degradation. Moreover, for larger models such as LLaVA-Next 13B and LLaVA-V1.5 13B, the performance loss after compression is even smaller.

Table 5: Performance (accuracy %) of compressed large vision-language models on the ScienceQA task under different compression ratios.

| Ratio | LLaVA-Next 7B | | LLaVA-Next 13B | | LLaVA-V1.5 13B | |
|---|---|---|---|---|---|---|
| | Ours | Standard SVD | Ours | Standard SVD | Ours | Standard SVD |
| 0.0 | 67.13 | 67.13 | 73.72 | 73.72 | 71.98 | 71.98 |
| 0.1 | 65.29 | 35.05 | 73.36 | 57.36 | 71.14 | 56.37 |
| 0.2 | 61.42 | 16.70 | 72.93 | 53.54 | 70.74 | 47.99 |
| 0.3 | 60.13 | 11.11 | 72.58 | 44.07 | 70.15 | 45.71 |

## I    IMPACT OF CALIBRATION SET SELECTION

This subsection examines the impact of calibration data used for the our method. Figure 7 illustrates the performance of compressed Vicuna-7B with different calibration datasets. The horizontal axis indicates the test set, while the vertical axis corresponds to the calibration set used during compression. The results show a clear trend: performance is best when the calibration set aligns with the test distribution. This pattern holds consistently across different compression ratios and supports the intuitive expectation that matching distributions reduce activation shift and improve whitening effectiveness. Conversely, domain mismatch between calibration and evaluation data leads to noticeable performance drops, especially at higher compression rates. These findings highlight the importance of choosing representative calibration data for distribution-aware methods. While we follow ASVD (Yuan et al., 2023) and adopt a 256-sample subset from WikiText-2 for most of our main experiments to ensure fair comparison, exploring more principled or adaptive calibration strategies remains a valuable direction for future work.

## J    COMPARISON OF COMPRESSED MODELS IN ZERO-SHOT SETTINGS

We conduct experiments on 16 representative subtasks from the **Massive Multitask Language Understanding (MMLU)** benchmark (Hendrycks et al., 2021), which covers a diverse set of knowl-

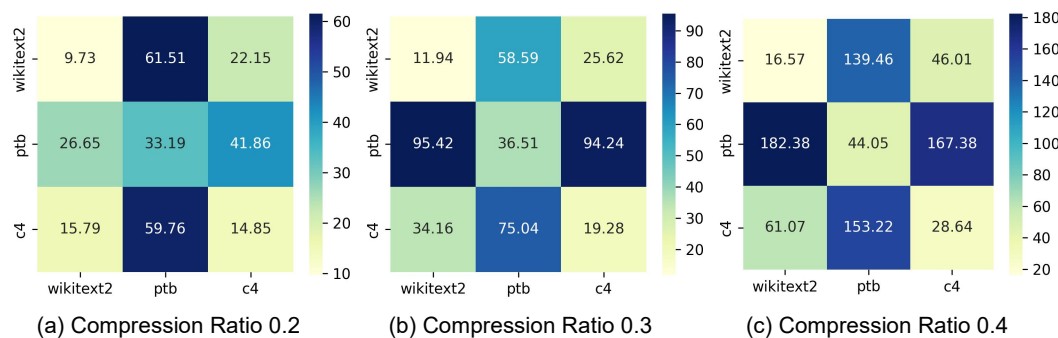

Figure 7: Perplexity with different calibration datasets.

edge domains including humanities, social sciences, STEM, and other disciplines. To make the results table concise and readable, we use the following abbreviations for subtask names:

- **HS_US_Hist**: High School US History
- **HS_World_Hist**: High School World History
- **Intl_Law**: International Law
- **Log_Fallacies**: Logical Fallacies
- **Bus_Ethics**: Business Ethics
- **Global_Facts**: Global Facts
- **Marketing**: Marketing
- **Virology**: Virology
- **HS_Gov_Pol**: High School Government and Politics
- **Human_Sexuality**: Human Sexuality
- **Prof_Psych**: Professional Psychology
- **US_Foreign_Pol**: US Foreign Policy
- **College_Bio**: College Biology
- **College_Phys**: College Physics
- **Comp_Sec**: Computer Security
- **Elec_Eng**: Electrical Engineering

Table 6 compares the zero-shot performance at a compression ratio of 20% between our proposed method and SVD-LLM, which represents the current state-of-the-art in SVD-based model compression. Accuracy is reported across three model backbones: Vicuna-V1.5-7B, Qwen2-7B, and LLaMA3-8B, with the performance gain of our method shown alongside. Overall, our method achieves better performance than the SVD-LLM in the majority of evaluated subtasks. For example, on the *High School Government and Politics* subtask under the Qwen2-7B model, our method improves accuracy from 24.35% to 41.97%, achieving a gain of +17.62%. Similarly, in the *Human Sexuality* subtask, accuracy increases from 24.43% to 39.69%, resulting in a gain of +15.26%. These substantial improvements across different domains and models indicate that our method generalizes well and provides a more robust alternative to traditional SVD-based compression strategies.

## K    ADDITIONAL RESULTS FOR LIGHTWEIGHT POST-COMPRESSION FINE-TUNING

Lightweight Post-Compression Fine-Tuning is a widely used approach (Ma et al., 2023; Wang et al., 2025b) for recovering performance after model compression. Due to page limitations, only a subset of the experimental results is presented in the main text; here we provide additional data in Table

Table 6: Comparison of accuracy between SVD-LLM and our method across diverse subtasks in the Massive Multitask Language Understanding Benchmark.

| Category | Subtask | Vicuna-V1.5-7B | | | Qwen2-7B | | | LLaMA3-8B | | |
|---|---|---|---|---|---|---|---|---|---|---|
| | | SVD-LLM | Ours | Gain | SVD-LLM | Ours | Gain | SVD-LLM | Ours | Gain |
| Humanities | HS_US_Hist | 0.3529 | 0.4167 | 0.0638 | 0.3039 | 0.3873 | 0.0834 | 0.2843 | 0.2892 | 0.0049 |
| | HS_World_Hist | 0.3882 | 0.4473 | 0.0591 | 0.3418 | 0.4641 | 0.1223 | 0.2405 | 0.2785 | 0.0380 |
| | Intl_Law | 0.3884 | 0.4711 | 0.0827 | 0.2810 | 0.3140 | 0.0330 | 0.3802 | 0.4876 | 0.1074 |
| | Log_Fallacies | 0.2822 | 0.3620 | 0.0798 | 0.2699 | 0.3558 | 0.0859 | 0.2699 | 0.3129 | 0.0430 |
| Other | Bus_Ethics | 0.3200 | 0.3600 | 0.0400 | 0.2800 | 0.4000 | 0.1200 | 0.2700 | 0.3100 | 0.0400 |
| | Global_Facts | 0.2400 | 0.3500 | 0.1100 | 0.1900 | 0.2200 | 0.0300 | 0.3500 | 0.3400 | -0.0100 |
| | Marketing | 0.3376 | 0.4530 | 0.1154 | 0.3547 | 0.4530 | 0.0983 | 0.3077 | 0.2991 | -0.0086 |
| | Virology | 0.3072 | 0.3614 | 0.0542 | 0.2651 | 0.2952 | 0.0301 | 0.2229 | 0.2349 | -0.0120 |
| Social science | HS_Gov_Pol | 0.3420 | 0.3834 | 0.0414 | 0.2435 | 0.4197 | 0.1762 | 0.2280 | 0.3057 | 0.0777 |
| | Human_Sexuality | 0.3588 | 0.4046 | 0.0458 | 0.2443 | 0.3969 | 0.1526 | 0.2214 | 0.2366 | 0.0152 |
| | Prof_Psych | 0.2827 | 0.3072 | 0.0245 | 0.2843 | 0.3268 | 0.0425 | 0.2810 | 0.2810 | 0.0000 |
| | US_Foreign_Pol | 0.4200 | 0.5200 | 0.1000 | 0.3000 | 0.4600 | 0.1600 | 0.2700 | 0.3500 | 0.0800 |
| STEM | College_Bio | 0.2708 | 0.3056 | 0.0348 | 0.2639 | 0.3472 | 0.0833 | 0.2917 | 0.2431 | -0.0486 |
| | College_Phys | 0.2941 | 0.3137 | 0.0196 | 0.2353 | 0.3039 | 0.0686 | 0.2255 | 0.2549 | 0.0294 |
| | Comp_Sec | 0.3200 | 0.3500 | 0.0300 | 0.2600 | 0.3200 | 0.0600 | 0.2900 | 0.3800 | 0.0900 |
| | Elec_Eng | 0.2897 | 0.3241 | 0.0344 | 0.2621 | 0.2759 | 0.0138 | 0.2897 | 0.3034 | 0.0137 |
| **Average** | | 0.3247 | 0.3831 | 0.0585 | 0.2737 | 0.3587 | 0.0850 | 0.2764 | 0.3067 | 0.0303 |

7. In most cases, our method outperforms the baseline SVD-LLM across different language models after lightweight post-compression fine-tuning. For example, at a compression ratio of 0.3, with a full update on the LLaMA3-8B model, the perplexity of our method on the PTB dataset is 27.5872, which is 6.5991 points lower than that of SVD-LLM (34.1863). As the compression ratio increases, this performance improvement becomes even more pronounced.

Table 7: Perplexity of our method across three language models under different compression ratios. After lightweight post-compression fine-tuning, our proposed method outperforms the baseline method SVD-LLM in most cases.

| Ratio | Method | Vicuna-7B | | | Qwen2-7B | | | LLaMA3-8B | | |
|---|---|---|---|---|---|---|---|---|---|---|
| | | WikiText-2 | PTB | C4 | WikiText-2 | PTB | C4 | WikiText-2 | PTB | C4 |
| 0.3 | SVD-LLM Partial Update | 9.5785 | 54.2648 | **16.0825** | 11.3648 | 33.1960 | 26.6838 | **14.5375** | 42.4973 | 38.2274 |
| | Ours Partial Update | **9.5781** | **53.7169** | 16.2617 | **11.2563** | **32.7206** | **26.5142** | 15.8428 | **31.0731** | **33.1449** |
| | SVD-LLM Full Update | 9.1292 | 52.7985 | **14.6065** | 10.2916 | 25.4728 | **20.7969** | 13.1556 | 34.1863 | 32.8736 |
| | Ours Full Update | **9.0960** | **52.1565** | 14.7191 | **10.1894** | **25.2227** | 21.3162 | 14.8881 | **27.5872** | **31.0844** |
| 0.4 | SVD-LLM Partial Update | 11.3849 | 77.1259 | 19.8853 | 16.1168 | 50.0841 | 37.0297 | **20.7856** | 74.2456 | 64.5032 |
| | Ours Partial Update | **11.3136** | **71.2715** | **19.8202** | **15.2767** | **44.3175** | **34.2671** | 22.2111 | **37.9897** | **49.5095** |
| | SVD-LLM Full Update | 10.5504 | 79.9615 | 17.3544 | 12.2817 | 34.9910 | 26.9585 | **17.7929** | 51.4241 | 55.1955 |
| | Ours Full Update | **10.5386** | **65.2732** | **17.3092** | **12.2373** | **32.5209** | **26.6492** | 20.4144 | **34.3254** | **43.0600** |
| 0.5 | SVD-LLM Partial Update | 14.7036 | 130.6588 | 26.8014 | 22.4540 | 74.3111 | 54.7868 | 51.7419 | 193.0543 | 124.1097 |
| | Ours Partial Update | **14.3285** | **108.6498** | **25.6585** | **20.2631** | **58.0048** | **46.4589** | **36.6911** | **75.2853** | **73.5259** |
| | SVD-LLM Full Update | 12.7389 | 180.4358 | 21.5852 | 16.3075 | 46.8237 | 39.3111 | 49.4664 | 128.2454 | 85.4848 |
| | Ours Full Update | **12.6762** | **124.4990** | **21.5079** | **16.0868** | **40.7479** | **35.7692** | **32.1073** | **64.5531** | **64.6075** |

## L    ADDITIONAL CONVERGENCE ANALYSIS OF LoRA INITIALIZATION STRATEGIES

Figure. 8 presents training convergence curves of the three LoRA initialization methods under various rank settings on mathematical (MetaMathQA) and programming (Code Feedback) tasks. The results further confirm the advantage of our proposed initialization in accelerating convergence and improving training stability.

## M    SUITABILITY OF SVD-LLM FOR PEFT INITIALIZATION

This section clarifies that distribution-aware low-rank approximations serve as a stronger initialization for LoRA-style PEFT than existing SVD-based methods.

Why distribution-aware initialization helps: Recent work such as PiSSA has highlighted that an initialization which better preserves the model's task-relevant subspace provides a superior starting

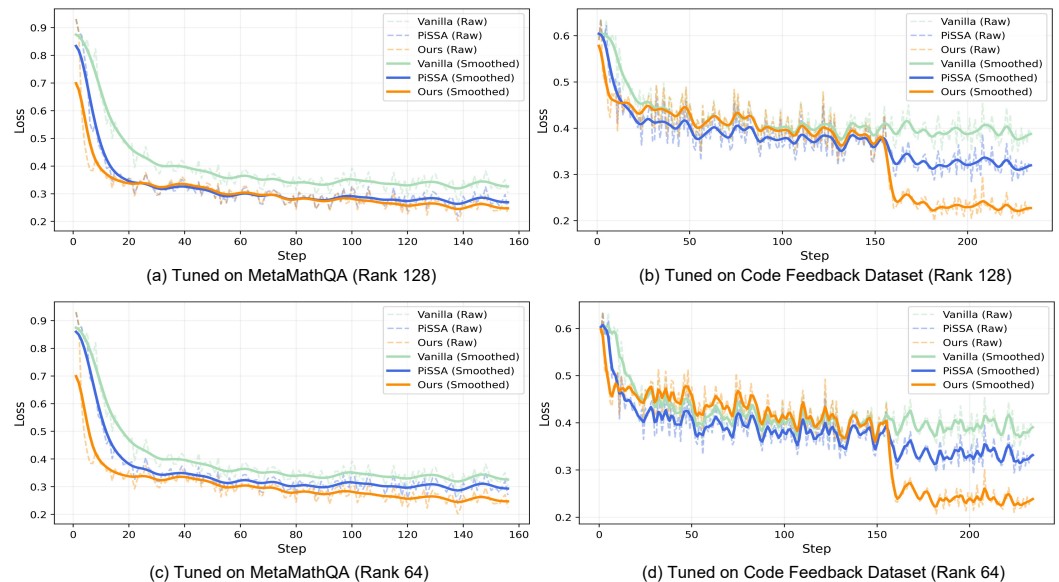

Figure 8: Training loss for fine-tuning Vicuna-V1.5-7B with LoRA (rank 128 & 64) under three different initialization schemes: Vanilla Kaiming Initialization, PiSSA, and our proposed method. (a) Results on the MetaMathQA dataset; (b) results on the Code Feedback dataset. Dashed lines represent raw training loss values, while solid lines denote Gaussian-smoothed curves for improved visual clarity.

point for PEFT. Concretely, distribution-aware whitening reduces the cumulative, layer-wise distortion introduced by low-rank approximation, so the removed components have smaller impact on model outputs. As a result, when used as an initialization for LoRA-style training, such approximations: 1)Start closer to a good local optimum. 2)Typically converge faster and achieve a better final solution.

Why SVD-LLM may be insufficient: Methods like SVD-LLM overlook approximation-induced distribution shifts across layers. When the retained rank is small, approximation errors accumulate, and the preserved subspace may fail to capture crucial predictive directions. Therefore, while SVD-LLM can improve over vanilla LoRA in some settings, it does not consistently match the benefits offered by distribution-aware initialization.

Empirical evidence: We evaluated Vicuna-7B across a set of math and coding tasks. Figure 9 reports results (higher is better) for different initializations and rank settings, averaged across 3 random seeds. These results demonstrate that: SVD-LLM can outperform vanilla LoRA. However, SVD-LLM is not consistently better than PiSSA. Our distribution-aware initialization yields the most consistent improvements across tasks and rank settings.

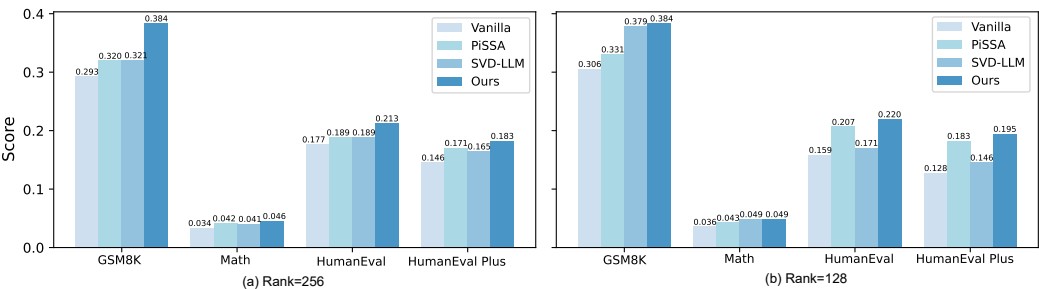

Figure 9: Performance comparison of four PEFT initialization methods on Vicuna-7B across four tasks for both rank 256 and 128.

# N  IMPACT OF PEFT ON OTHER TASKS AFTER FINE-TUNING ON SPECIFIC TASKS

Both our method and PiSSA initialize the LoRA modules using the important components of the model's weight matrices. A natural question arises: does this initialization lead to severe forgetting of previously learned knowledge? Table 8 compares the performance of the Qwen2-7B model fine-tuned on mathematical and coding tasks with a rank of 256, evaluated on the MMLU Benchmark. Interestingly, models fine-tuned using different initialization strategies show almost no degradation in performance compared to the original model. In some cases, slight improvements are even observed. These results demonstrate that initializing LoRA with the most important components of the weight matrices does not cause severe forgetting, indicating good usability and robustness of our approach.

Table 8: Performance of models fine-tuned on different tasks using various LoRA initialization methods, evaluated on the MMLU Benchmark. The results show that our method does not cause severe forgetting.

| Category | Subtask | Original | Peft on MetaMathQA | | | Peft on Code Feedback | | |
| --- | --- | --- | --- | --- | --- | --- | --- | --- |
| | | | Vanilla | PiSSA | Ours | Vanilla | PiSSA | Ours |
| Humanities | HS_US_Hist | 0.8431 | 0.8529 | 0.8284 | 0.8529 | 0.8431 | 0.8480 | 0.8431 |
| | HS_World_Hist | 0.8312 | 0.8354 | 0.8397 | 0.8312 | 0.8354 | 0.8439 | 0.8397 |
| | Intl_Law | 0.8182 | 0.8512 | 0.8347 | 0.8512 | 0.8512 | 0.8347 | 0.8512 |
| | Log_Fallacies | 0.8037 | 0.8037 | 0.7362 | 0.8221 | 0.7791 | 0.7853 | 0.7853 |
| Other | Bus_Ethics | 0.7800 | 0.7600 | 0.7700 | 0.7800 | 0.7900 | 0.7800 | 0.7700 |
| | Global_Facts | 0.4900 | 0.4900 | 0.2800 | 0.4900 | 0.4900 | 0.4800 | 0.4900 |
| | Marketing | 0.9274 | 0.9231 | 0.8846 | 0.9188 | 0.9274 | 0.9274 | 0.9188 |
| | Virology | 0.5301 | 0.5422 | 0.5482 | 0.5422 | 0.5422 | 0.5422 | 0.5482 |
| Social science | HS_Gov_Pol | 0.9016 | 0.9067 | 0.9016 | 0.9067 | 0.8964 | 0.9067 | 0.9067 |
| | Human_Sexuality | 0.8321 | 0.8473 | 0.8244 | 0.8397 | 0.8397 | 0.8321 | 0.8321 |
| | Prof_Psych | 0.7484 | 0.7500 | 0.7533 | 0.7614 | 0.7533 | 0.7533 | 0.7565 |
| | US_Foreign_Pol | 0.8800 | 0.9000 | 0.8900 | 0.9000 | 0.8900 | 0.8900 | 0.8900 |
| Stem | College_Bio | 0.8056 | 0.7917 | 0.7847 | 0.7917 | 0.7917 | 0.7986 | 0.7847 |
| | College_Phys | 0.4412 | 0.4216 | 0.3725 | 0.4020 | 0.4118 | 0.3922 | 0.4510 |
| | Comp_Sec | 0.7700 | 0.7800 | 0.7800 | 0.7800 | 0.7800 | 0.7800 | 0.7800 |
| | Elec_Eng | 0.7241 | 0.7448 | 0.6828 | 0.7310 | 0.7379 | 0.7241 | 0.7310 |
| **Average** | | 0.7579 | 0.7625 | 0.7319 | 0.7626 | 0.7600 | 0.7574 | 0.7611 |

