# OpenReview forum: "Mitigating Error Propagation in Low-Rank Approximation of Large Models via Distribution-Aware Whitening"
_ICLR.cc/2026/Conference — Submitted to ICLR 2026_

### Official Review · Reviewer_i2oL · 2025-10-22

**Soundness:** 2
**Presentation:** 1
**Contribution:** 2
**Rating:** 2
**Confidence:** 3

**Summary:**

The paper proposed a distribution-aware whitening framework for low-rank approximation of LLMs. The key idea is: when applying low-rank factorization/compression of model weights or fine-tuning via low-rank adapters, the input feature distributions at each layer may shift and become anisotropic, which can amplify approximation error across layers.

**Strengths:**

The problem addressed is relevant and timely: low-rank approximation and efficient fine‐tuning of large models is important for deployment.

**Weaknesses:**

1 The core method, i.e., how exactly the whitening is applied, over what time/blocks, how the low‐rank approximation is integrated, how the discarded components are selected, is not clearly or cleanly presented.
2 It is not entirely clear how this method compares to other normalization or whitening‐inspired compression methods in literature; the novelty relative to prior work

**Questions:**

Have you compared to simpler baselines such as applying a standard normalization before low-rank factorization?

How much marginal benefit does your whitening bring over such simpler normalization?

---

> ### Author Response · Authors · 2025-11-22
> **Response to Reviewer i2oL**
>
> We greatly appreciate the time and effort you have devoted to reviewing our work. We hope that our detailed responses will clarify your questions and concerns, and we would be happy to provide further explanations if needed.
>
> > Q1：*Concern on Method Clarity.*
>
> R1: We thank the reviewer for the comment. We would like to clarify the presentation and key steps of our method to address the concern.
>
> - Motivation and problem analysis: In Section 3.1, we analyze why directly applying SVD for model compression can lead to substantial information loss. This motivates the need for a more careful treatment of feature distributions during low-rank approximation.
> - Distribution-aware whitening and component selection: In Section 3.2, we introduce whitening of the input features for each weight matrix, formalized in Equations (3–5). Importantly, this whitening is **different from standard data normalization or PCA whitening** used in standard data preprocessing. The goal here is to **establish a direct link between truncation (rank selection) and output distortion**, enabling us to rank the components and choose which ones to discard systematically. At the end of Section 3.2, we further describe how this method can be applied for both post-training compression and PEFT initialization.
> - Theoretical justification: In Section 3.3, we analyze how low-rank approximation errors propagate across layers and show theoretically why distribution-aware whitening mitigates such errors. By explicitly aligning the retained components with the model’s fidelity, our method ensures that truncation has minimal adverse effects on model predictions. Simple operations such as weight normalization alone cannot achieve this objective.
> - Regarding which layers are compressed, we follow existing works: all layers are uniformly compressed by α to achieve an overall compression of α. Only in the Subsection discussing dynamic rank allocation do we apply different compression ratios across layers, while keeping the total number of compressed parameters equal to α. For clarity, we have added corresponding explanations in the experimental section, as suggested.
>
> Overall, our method combines theoretical analysis with practical guidelines for implementation, which is further validated in experiments showing clear empirical improvements over baselines.
>
> > Q2: *It is not entirely clear how this method compares to other normalization or whitening‐inspired compression methods in literature.*
>
> A2: Thanks for your comments. In the Related Work section (Subsection 2.1), we review prior studies on low-rank approximation in model compression and point out that these methods often overlook the feature distribution shifts introduced during compression, which can amplify error propagation. To address this issue, our method incorporates a distribution-aware whitening mechanism that effectively mitigates distribution drift, thereby enhancing the stability and robustness of low-rank compression.
>
> In the Analysis section (Section 3.3), we further compare our approach with the whitening strategies used in existing methods and analyze their respective impacts on error propagation. In the Experiments section, extensive evaluations on 10+ different large-scale models demonstrate that our method consistently outperforms existing approaches across a wide range of settings.
>
> > Q3: Have you compared to simpler baselines such as applying a standard normalization before low-rank factorization? How much marginal benefit does your whitening bring over such simpler normalization?
>
> A3: We thank the reviewer for this question. Our approach is fundamentally different from simply applying a standard normalization before low-rank factorization: the proposed distribution-aware whitening is specifically designed to align truncation with output distortion and to suppress cross-layer error amplification—capabilities that standard normalization does not provide. To directly address the reviewer’s concern, we implemented a baseline that applies standard normalization prior to SVD when compressing Vicuna-7B by 30%. This baseline produces a WikiText-2 perplexity of 112.86, far worse than the 11.94 achieved by our method under the same setting, demonstrating that simple normalization neither mitigates distribution shift nor preserves model fidelity.

---

### Official Review · Reviewer_8dzz · 2025-10-31

**Soundness:** 4
**Presentation:** 4
**Contribution:** 3
**Rating:** 8
**Confidence:** 3

**Summary:**

This paper tackles a core weakness of SVD-style low-rank compression and LoRA init—ignoring how upstream compression shifts activation distributions—by introducing a distribution-aware whitening framework that re-estimates each layer’s input covariance on the fly, whitens activations, performs SVD in the isotropic space, and maps the result back to the original space. The authors give a clean error-propagation analysis contrasting raw, static-whitening, and dynamic-whitening schemes, showing “whitening drift” can amplify distortion across depth and that distribution awareness tightens layerwise reconstruction bounds. Empirically, across several LLMs and datasets, the method yields lower perplexity than SVD-LLM under fixed and dynamically allocated compression ratios, is more robust when combined with quantization (e.g., GPTQ-4bit), and also serves as a stronger LoRA initialization than PiSSA, improving convergence and downstream scores. Overall, it’s a principled and practically relevant refinement of low-rank approximation that links compression fidelity to evolving feature statistics and demonstrates consistent gains without architectural changes.

**Strengths:**

### Strong theoretical grounding:
The paper provides a formal analysis of error propagation under different whitening schemes, clearly quantifying how distribution misalignment across layers amplifies reconstruction errors. This theoretical treatment makes the motivation and advantage of the proposed method well-justified rather than heuristic.

### Principled and general framework:
The proposed distribution-aware whitening is simple, modular, and architecture-agnostic. It can be used as a drop-in replacement for conventional SVD in model compression or as a better initialization strategy for LoRA without altering existing architectures or training pipelines.

### Comprehensive empirical validation:
The authors evaluate the method across multiple LLMs (e.g., LLaMA-7B, LLaMA3-8B, Qwen2-7B) and datasets (WikiText-2, PTB, C4) under various compression ratios. Results consistently show lower perplexity and better convergence than SVD-LLM and PiSSA, confirming both compression and fine-tuning benefits.

### Compatibility and robustness:
The approach complements other compression techniques like pruning and quantization rather than competing with them, and it demonstrates stability under lightweight post-compression fine-tuning.

**Weaknesses:**

While the paper provides strong theoretical and empirical validation in terms of perplexity and convergence, it lacks practical efficiency metrics. Since this is a model compression study, reporting inference throughput, latency, and actual memory usage compared to baselines (e.g., SVD-LLM, pruning-based methods) would provide a more comprehensive understanding of real-world benefits.

**Questions:**

1. Did the authors perform any retraining or fine-tuning after compression? It would be helpful to clarify whether the reported results are obtained purely from post-compression evaluation or involve additional training steps.

2. Since this work focuses on model compression, it would be informative to report inference efficiency metrics such as throughput  and GPU memory usage compared to baselines.

3. In Table 1, including the baseline performance at a compression ratio of 0% (i.e., the uncompressed model) would help readers better understand the degradation trend and relative impact of compression.

### Suggestions:

1. It would be interesting to compare a compressed large model (e.g., Qwen3-8B compressed to 50%) against a smaller pre-trained model of similar size (e.g., Qwen3-4B). If the compressed Qwen3-8B outperforms Qwen3-4B, it would suggest that training a larger model once and then applying model compression for different deployment scales could be more cost-effective than maintaining multiple model families. Such an analysis could significantly strengthen the paper’s practical value and relevance.


2. Extending the proposed distribution-aware whitening approach to vision models (e.g., Vision Transformer) would also be valuable.
Evaluating its applicability to visual backbones or multimodal architectures—such as MaskAlign [1] or PELA [2]—could demonstrate broader generality beyond language models.

---

[1] Xue et al. Stare at What You See: Masked Image Modeling Without Reconstruction. CVPR 2023.
[2] Guo et al. PELA: Learning Parameter-Efficient Models with Low-Rank Approximation. CVPR 2024.

---

> ### Author Response · Authors · 2025-11-22
> **Response to Reviewer 8dzz (Part 1)**
>
> We sincerely thank you for your time and thoughtful, detailed comments. Below, we provide our responses and would be happy to provide further clarifications if needed.
>
> > Q1:*Did the authors perform any retraining or fine-tuning after compression?*
>
> A1: We thank the reviewer for the helpful question. We have clarified this point more explicitly in the revised experimental section.
>
> Unless otherwise specified, **all reported main-text results are purely compression evaluations without any retraining or post-compression fine-tuning**. This is intentional: it allows us to directly compare how different low-rank approximation methods affect model quality in isolation.
>
> Due to page limits, our previous submission placed the analysis of post-compression fine-tuning results in Appendix H and Appendix I. These experiments demonstrate that our distribution-aware approach provides further strong improvements when lightweight fine-tuning is applied. In the revised version, we moved and expanded the corresponding discussion into the main text so that readers can more easily understand both evaluation settings.
>
> > Q2: *It would be informative to report inference efficiency metrics such as throughput and GPU memory usage.*
>
> A2: We appreciate the reviewer’s excellent suggestion. Adding inference efficiency metrics indeed improves both the clarity and practical relevance of our experimental section.
>
> In the revised version, we have included throughput and GPU memory usage (covering both weight memory and activation memory) under different output lengths and batch sizes. Since SVD-based compression methods share the same architecture and differ only in the number of retained parameters, we focus on presenting the impact of varying compression ratios rather than comparing across SVD variants. Below is an example of throughput and total memory usage for Vicuna-7B at output length 32 and batch size 128 under different compression ratios:
>
> Total Memory (GB) | Throughput (tokens/s) | Out_Len | Batch_Size | Ratio
> --- | --- | --- | --- | ---
> 34.09 | 641.51 | 32 | 128 | 0.0
> 29.37 | 734.28 | 32 | 128 | 0.2
> 26.81 | 762.21 | 32 | 128 | 0.3
> 24.66 | 870.80 | 32 | 128 | 0.4
> 22.03 | 923.00 | 32 | 128 | 0.5
>
> We further added corresponding bar charts and discussions in the revised experimental section to clearly illustrate (a) how inference latency changes after compression, (b) memory savings at different compression ratios, and (c) how inference throughput scales with model size.
>
> > Q3: *including the baseline performance of the uncompressed model would help readers better understand the degradation trend*
>
> A3: We thank the reviewer for the helpful suggestion. Following your recommendation, we have added the baseline (uncompressed) performance and visualized the degradation trend using additional line plots. Specifically, we compare our method with pruning-based baselines across compression ratios of 0%, 10%, 20%, 30%, 40%, and 50%.
> Given that Table 1 is already space-intensive, presenting the results as additional curves allows readers to more clearly observe how performance changes as compression increases. The new plots have been added to the revised version to improve readability and interpretability.
>
> > Q4: *It would be interesting to compare a compressed large model against a smaller pre-trained model of similar size.*
>
> We thank the reviewer for the thoughtful question. This is indeed an important direction and one of the long-term goals for model compression. For rigor, we performed the suggested comparison by compressing **Qwen2.5-14B → Qwen2.5-7B size** and **Vicuna-13B → Vicuna-7B size**, and evaluated both language modeling perplexity and downstream tasks. The results (shown below) indicate that although our method substantially improves over existing compression techniques, a compressed large model still does not fully match the performance of an original smaller model trained from scratch with its own capacity constraints.
>
> | Model                     | PPL  | MRPC  | RTE   | SST-2 |
> |---------------------------|------|-------|-------|-------|
> | Qwen2.5-14B (Compressed)  | 11.55 | 0.6838 | 0.6823 | 0.7890 |
> | Qwen2.5-7B (Original)       | 7.46 | 0.6677 | 0.8412 | 0.9358 |
> | Vicuna-13B (Compressed)   | 9.14 | 0.6838 | 0.5704 | 0.7878 |
> | Vicuna-7B (Original)        | 6.78 | 0.7157 | 0.6390 | 0.8073 |
>
> This performance gap is not unique to our approach: a native 7B model is trained from large-scale data and naturally learns to pack essential knowledge into its parameter budget, whereas compressing a 13B/14B model inevitably removes information, even with improved distribution-aware truncation.  Developing effective post-compression recovery or adaptation strategies can be an important and valuable research direction for future work.

---

> > ### Author Response · Authors · 2025-11-22
> > **Response to Reviewer 8dzz (Part 2)**
> >
> > > Q5: *Extending the proposed distribution-aware whitening approach to vision models would also be valuable.*
> >
> > A5: We thank the reviewer for the insightful suggestion. In response, we have extended our distribution-aware whitening approach to large vision-language models (VLMs), including **LLaVA-v1.5-13B**, **LLaVA-Next-7B**, and **LLaVA-Next-13B**. We evaluate on the **ScienceQA** task and compare our method with standard SVD-based compression.
> >
> > Our results demonstrate that the proposed method effectively generalizes to multi-modal models. For example, compressing LLaVA-Next-13B by 10% of parameters, our method leads to only 0.36% performance drop, whereas standard SVD causes a 16.36% drop. Similar trends are observed across the other models, indicating that distribution-aware whitening preserves critical information better than standard low-rank approximations.
> >
> > | Ratio | LLaVA-Next 7B |        | LLaVA-Next 13B |        | LLaVA-v1.5 13B |        |
> > |-------|---------------|--------|----------------|--------|----------------|--------|
> > |       | Ours      | Standard SVD | Ours      | Standard SVD | Ours      | Standard SVD |
> > | 0.0     | 67.13         | 67.13  | 73.72         | 73.72  | 71.98         | 71.98  |
> > | 0.1   | 65.29         | 35.05  | 73.36         | 57.36  | 71.14         | 56.37  |
> > | 0.2   | 61.42         | 16.70  | 72.93         | 53.54  | 70.74         | 47.99  |
> > | 0.3   | 60.13         | 11.11  | 72.58         | 44.07  | 70.15         | 45.71  |
> >
> > These findings suggest that our compression method has promising potential for multi-modal settings and motivate further exploration in vision-language models. We have added these results to the appendix for readers’ reference. Thank you for this nice suggestion again.

---

### Official Review · Reviewer_E5cT · 2025-11-01

**Soundness:** 2
**Presentation:** 3
**Contribution:** 3
**Rating:** 6
**Confidence:** 4

**Summary:**

This paper proposes a distribution-aware whitening framework for low-rank approximation in large language models. The core contribution is addressing the shifts in feature distributions induced by the approximation process, which can lead to error amplification across layers. The method dynamically whitens layer inputs based on evolving feature distributions to ensure second-order isotropy of input features, allowing discarded components in low-rank approximation to have minimal impact on model outputs. The authors provide theoretical analysis on how distribution misalignment leads to error propagation and demonstrate experimental results on various large language models for both post-training compression and as initialization for LoRA-style parameter-efficient fine-tuning.

**Strengths:**

1. **Important problem**: Addresses a genuine issue in low-rank approximation where distribution shifts can cause cumulative errors across layers. The observation that anisotropic input distributions can cause small singular values to align with high-energy input components (leading to disproportionate output distortion) is well-articulated and provides strong motivation for the proposed approach.
2. **Theoretical motivation**: Provides theoretical analysis linking distribution alignment to error propagation, which adds depth to the empirical work.  The decomposition of whitening drift into initial residual and compression-induced shift (Equation 12) offers valuable insights into the source of performance degradation.
3. **Clear presentation of the core idea**: The motivation and high-level approach are communicated effectively. The progression from problem identification to solution design is logical and well-structured.

**Weaknesses:**

1. **Some kind of Limited novelty in core technique:** The use of whitening transformations to decorrelate features is a well-established technique in machine learning. While the dynamic, layer-wise application is novel, the core mathematical framework (ZCA whitening via Cholesky decomposition) is standard. The main contribution appears to be the engineering insight of recomputing whitening based on compressed activations rather than a fundamentally new algorithmic approach.
2. **Computational overhead not thoroughly analyzed:** While Appendix A acknowledges additional computational cost from per-layer whitening operations, the paper lacks quantitative analysis of this overhead. Critical missing information includes: (a) wall-clock time comparison during compression, (b) memory overhead during the compression process, (c) impact on inference latency after compression, and (d) scalability analysis showing how overhead grows with model size. For practitioners, these practical considerations are as important as perplexity improvements.

**Questions:**

1. [**Theoretical guarantees:**]
    - Can you provide tighter bounds that incorporate the regularization parameter ε?
    - How do the Lipschitz constants ρₗ behave in practice for transformer layers?
2. **Scalability:** Your largest model is 13B parameters. Do you expect the method to scale to 70B+ models?
3. **Dynamic compression:** In Section 4.4, you combine your method with dynamic ratio allocation. How does layer-wise variation in compression ratio affect the whitening computation? Should the whitening strategy be different for layers with different compression ratios?

---

> ### Author Response · Authors · 2025-11-22
> **Response to Reviewer E5cT (Part 1)**
>
> We sincerely thank you for your time and thoughtful, detailed comments. Below, we provide our responses and would be happy to provide further clarifications if needed.
>
> > Q1：*Some kind of Limited novelty in core technique：the core mathematical framework (ZCA whitening via Cholesky decomposition) is standard.*
>
> A1:We thank the reviewer for this comment. We agree that the underlying mathematical tools (e.g., whitening via Cholesky decomposition) are standard. However, our contribution does not lie in proposing a new whitening algorithm, but in introducing a new analysis of how feature distribution isotropy relates to low-rank truncation error in LLMs, and in establishing a principled connection between truncation and output distortion.
>
> Specifically:
>
> • Prior work has not analyzed how anisotropy in layers input leads to cross-layer error amplification.
>
> • We introduce a simple yet effective framework that links the input covariance structure to output distortion, thereby aligning the retained low-rank components with model's output fidelity, which is different from standard whitening used in data preprocessing.
>
> • The proposed formulation leads to substantial empirical improvements across 10+ LLMs under diverse settings, consistently outperforming existing SVD-based compression baselines.
>
> Thus, while the mathematical building blocks are classical, the problem formulation, the role whitening plays within the compression pipeline, and the theoretical connections we establish are novel. The contribution of our work lies in showing how distribution-aware whitening can significantly reduce low-rank distortion in LLMs—a perspective has not been well explored in prior compression and PEFT literature. We hope this provides a fresh and useful angle for understanding low-rank model compression.
>
> > Q2: *Computational overhead not thoroughly analyzed.*
>
> A2: We appreciate the reviewer’s excellent suggestion. The computational overhead of whitening is indeed an important practical factor, and we have expanded our analysis accordingly.
>
> First, regarding the compression-time overhead: on an NVIDIA A100 GPU, compressing Vicuna-7B by 30% using standard SVD takes approximately 12 minutes, while incorporating our per-layer whitening operations takes around 25 minutes. While this adds some overhead during compression, it is a one-time offline process and the performance gains justify the modest increase.
>
> Second, in the revised manuscript, we provide a quantitative analysis of inference efficiency, reporting throughput and GPU memory usage (covering both weight memory and activation memory) under different output lengths and batch sizes. Below is an example of throughput and total memory usage for Vicuna-7B at output length 32 and batch size 128 under different compression ratios (Ratio 0.0 denotes the original model without compression):
>
> Total Memory (GB) | Throughput (tokens/s) | Out_Len | Batch_Size | Ratio
> --- | --- | --- | --- | ---
> 34.09 | 641.51 | 32 | 128 | 0.0
> 29.37 | 734.28 | 32 | 128 | 0.2
> 26.81 | 762.21 | 32 | 128 | 0.3
> 24.66 | 870.80 | 32 | 128 | 0.4
> 22.03 | 923.00 | 32 | 128 | 0.5
>
> We further added corresponding bar charts and discussions in the revised experimental section to clearly illustrate (a) how inference latency changes after compression, (b) memory savings at different compression ratios, and (c) how inference throughput scales with model size. We hope these additions address the reviewer’s concerns and provide a more complete reference for readers.
>
> >Q3.1: Can you provide tighter bounds that incorporate the regularization parameter $\epsilon$?
>
> A3.1:We thank the reviewer for the insightful question regarding the regularization parameter $\epsilon$.
>
> In our analysis, $\epsilon$ appears in the static whitening matrix $S_\ell^{\mathrm{stat}} = (X_{\ell-1} X_{\ell-1}^\top + \epsilon I)^{-1/2},$ primarily for numerical stability. Its contribution to the initial whitening residual can be explicitly bounded as
>
> $\delta_\ell^{\mathrm{init}} = \left\| ( \Sigma_{\ell-1} + \epsilon I)^{-1/2} \Sigma_{\ell-1} (\Sigma_{\ell-1} + \epsilon I)^{-1/2} - I \right\|_2$
>
> $\ \ \ \ \ \ \le \frac{\epsilon}{\lambda_{\min}(\Sigma_{\ell-1})},$
>
> where $\Sigma_{\ell-1}$ is the input covariance before compression. This provides a tight, quantitative measure of how $\epsilon$ affects the deviation from ideal whitening. In practice, $\epsilon$ is chosen to be very small (e.g., $10^{-5}$), so its contribution is typically negligible compared with the distribution shift caused by upstream compression. Nonetheless, this explicit bound allows readers to reason about the trade-off between numerical stability and approximation tightness.

---

> > ### Author Response · Authors · 2025-11-22
> > **Response to Reviewer E5cT (Part 2)**
> >
> > >Q3.2: How do the Lipschitz constants ρₗ behave in practice for transformer layers?
> >
> > A3.2:
> > Thank you for this insightful question. We measured the Lipschitz constants for the query, key, and value projections in the attention sublayers, as well as for the two linear layers in the feed-forward sublayers. The results are visualized in Appendix E.
> >
> > Empirically, we observe the following trends across LLaMA-7B, Vicuna-7B, Qwen2-7B, and LLaMA3-8B:
> >
> > - In attention sublayers, the query and key projections (Q/K) generally have higher Lipschitz constants than the value projections (V), reflecting their greater sensitivity to input perturbations.
> > - In feed-forward sublayers, the first linear layer (W1) tends to have larger Lipschitz constants than the second (W2).
> > - In deeper layers, particularly near the output, Lipschitz constants tend to increase slightly.
> >
> > These Lipschitz constants indicate that small errors can be amplified through the network, potentially harming model fidelity. Our distribution-aware whitening mechanism helps suppress such error propagation during low-rank compression.
> >
> > > Q4: *Your largest model is 13B parameters. Do you expect the method to scale to 70B+ models?*
> >
> > A4: We thank the reviewer for this important question. The core contribution of our work is the analysis of error propagation during low-rank compression and the proposal of a principled approach to mitigate distortion. This approach is fundamentally model-size agnostic and should generalize to much larger models.
> >
> > While computational resources limited our ability to experiment on 70B+ models, we additionally evaluated larger models, Qwen2.5-14B and Qwen2.5-32B, to verify scalability. As shown in the table below, our distribution-aware whitening consistently outperforms existing SVD-based compression across multiple compression ratios and datasets, demonstrating its effectiveness at scale. The numbers in parentheses correspond to compression ratios. Lower value indicates better performance.
> >
> > | Model         | Method  | WikiText-2 (0.3/0.4/0.5) | PTB (0.3/0.4/0.5) | C4 (0.3/0.4/0.5) |
> > |---------------|---------|---------------------------|------------------|-----------------|
> > | Qwen2.5-14B   | SVDLLM  | 11.8759 / 20.3064 / 62.1082 | 92.8457 / 279.7561 / 919.9447 | 56.8987 / 147.1329 / 384.9623 |
> > |               | Ours    | 11.0161 / 14.8418 / 23.1239 | 69.0246 / 121.1008 / 210.0547 | 44.9981 / 80.0062 / 149.4119 |
> > | Qwen2.5-32B   | SVDLLM  | 30.962 / 38.669 / 46.788 | 681.311 / 780.162 / 1196.067 | 344.032 / 381.856 / 483.195 |
> > |               | Ours    | 19.914 / 22.785 / 30.869 | 258.474 / 302.2099 / 413.135 | 224.699 / 286.357 / 379.361 |
> >
> > > Q5:*Should the whitening strategy be different for layers with different compression ratios?*
> >
> > A5:We thank the reviewer for this good question. Our work analyzes the propagation of compression-induced errors and establishes a principled link between truncation and output distortion to improve compression performance. Specifically, the whitening operation helps rank eigencomponents based on their expected impact on model outputs. This procedure is independent of the layer-wise compression ratio: whether a uniform or dynamic rank allocation is applied, the whitening computation remains the same.

---

> > > ### Comment · Reviewer_E5cT · 2025-11-26
> > > **Thanks for your explanation. I decide to keep the rating.**
> > >
> > > Thank you for providing a comprehensive and detailed explanation, which has effectively addressed all my inquiries. I am particularly looking forward to the public release of your reproducible GitHub repository.

---

> > > > ### Author Response · Authors · 2025-11-26
> > > >
> > > > Thank you for your encouraging comments. We are pleased that our responses have satisfactorily addressed your concerns. We will make the code reproducible for future researchers to reference and hope to contribute to the development of the community.

---

### Official Review · Reviewer_xHtd · 2025-11-02

**Soundness:** 3
**Presentation:** 3
**Contribution:** 1
**Rating:** 2
**Confidence:** 4

**Summary:**

Improving over the earlier methods of low rank decomposition, the authors propose that we should incorporate the compression of earlier layers to recompute the input distributions for current layers since injected error at earlier layers affects the input distributions. They theoretically analyze this error. This idea can be extended to model compression as well as PEFT initialization.

**Strengths:**

The paper is well written, self-contained (for a person like me who does not keep up with latest in model compression) and easy to read. It identifies and solves an important ignored issue in previous methods.

**Weaknesses:**

1. I do not buy the idea that low-rank is a promising model compression method (post training compression). For instance, the perplexity values in table 1 essentially say that in most cases model is not useful at all after compression. Is there a reason why authors believe that low-rank is a promising method? Related to this, there are improvements in table 1 for sure with using adjustment proposed. But the absolute values are too large for this table to be meaningful in my opnion.

2. The idea is important, self contained and potentially impactful. However, it is incremental since whitening is a generally established idea. I am not sure if the contribution is justifies an paper.

3. What happens if you only use SVD-LLM whitening (without adjustment for previous layers) in PEFT ?  I believe the impact of adjusting for previous errors will be minimal when used only for initialization.

**Questions:**

see weaknesses

---

> ### Author Response · Authors · 2025-11-22
> **Response to Reviewer xHtd (Part 1)**
>
> We greatly appreciate the time and effort you have devoted to reviewing our work. We hope that our detailed responses will clarify your questions and concerns, and we would be happy to provide further explanations if needed.
>
> > Question1:*“I do not buy the idea that low-rank is a promising post-training compression method… the perplexity values in Table 1 are too high… the absolute numbers make the table not meaningful.”*
>
> Response1:
> We thank the reviewer for raising this insightful concern, which helps us improve the clarity and presentation of our work. We address it from three perspectives.
>
> **1）Low-rank approximation remains an actively used and promising direction**
>
> Recent work shows that low-rank approximations continues to play an important role in both post-training compression (e.g., AWSVD, KSVD, SVD-LLM, DipSVD) and parameter-efficient fine-tuning (e.g., LoRA, PiSSA). These methods remain widely adopted in practice because low-rank approximations offer several unique advantages. For example, they preserve architectural compatibility and enable structured, hardware-friendly decompositions.
>
> Our contribution does not aim to claim that ​*low-rank is the best compression method*​, but rather to show that ​**existing low-rank methods suffer from a overlooked issue related to distribution shift, and that properly addressing this issue significantly improves their usability and stability**​. This is precisely where our work provides substantial benefits.
>
> **2）On the high perplexity values in Table 1**
>
> We agree that the absolute perplexity values in Table 1 are high. This is ​*expected by design*​:
> Table 1 reports ​**pure post-training low-rank approximation without any recovery techniques**​, in order to isolate and directly evaluate the intrinsic distortion introduced by different low-rank formulations.
>
> However, following standard practice in pruning-based compression, Lightweight Post-Compression Fine-Tuning can effectively recover model performance and reduce perplexity to a practically meaningful range. Moreover, as shown in Appendix, after lightweight fine-tuning, for instance, ​**the perplexity of Vicuna-7B with 50% compression ratio decreases from 26.51 to 12.67 on WikiText2, and from 118.67 to 21.50 on C4**​.
>
> **3）Our goal is not to compare low-rank vs. other compression families, but to improve the low-rank class itself**
>
> Our work focuses on improving the performance of low-rank approximations by addressing the overlooked issue of ​distribution misalignment. In Appendix H (​*Comparison with Pruning-based Approaches*​), we further show that low-rank methods retain advantages in certain scenarios compared with pruning-based methods, and our proposed approach remains compatible with existing quantization techniques.
>
> We appreciate the reviewer’s comment, which helped us refine the presentation. We have incorporated the discussion of these results into the main paper.
>
>
> > Question2: *The idea is important, self contained and potentially impactful. However, it is incremental since whitening is a generally established idea.*
>
> Response2:
> We thank the reviewer for recognizing the importance of our idea. We would like to clarify the key novelty of our approach.
>
> Our work is not merely about applying SVD or whitening to neural networks. Instead, we introduce a new perspective for understanding distortions caused by the approximation process. Specifically, we:
>
> 1. Model the error propagation during low-rank compression,
> 2. Analyze how layerwise distribution shifts  amplify such error propagation,
> 3. Propose distribution-aware whitening to enforce second-order isotropy, thereby establishing a correspondence between discarded low-rank components and their impact on model predictions.
>
> While the resulting method may appear straightforward at first glance, its theoretically driven simplicity is an advantage. Our framework:
>
> * Significantly mitigates cumulative layerwise distortion introduced by low-rank approximation in both post-training compression and PEFT,
> * Achieves performance comparable to widely used compression methods such as pruning,
> * And can be seamlessly combined with other model compression techniques, including quantization.
>
> We appreciate the reviewer’s acknowledgment of our idea and hope that this clarifies the novelty and significance of our approach.

---

> > ### Author Response · Authors · 2025-11-22
> > **Response to Reviewer xHtd (Part 2)**
> >
> > > Question3：*What happens if you only use SVD-LLM whitening (without adjustment for previous layers) in PEFT ?*
> >
> > Response3:
> >
> > We thank the reviewer for this question. Below we clarify our rationale and provide empirical evidence showing that *distribution-aware* low-rank approximations serve as a stronger initialization for LoRA-style PEFT than naive SVD-based method.
> >
> > **Why distribution-aware initialization helps:** It's highlighted in the recent proposed method PiSSA that an initialization which better preserves the model’s *task-relevant* subspace provides a superior starting point for PEFT. Concretely, distribution-aware whitening reduces the cumulative, layerwise distortion introduced by low-rank approximation, so the removed components are those with smaller impact on model outputs. As a result, when used as an initialization for LoRA-style training, such approximations (i) start closer to a good local optimum and (ii) thus typically converge faster and to a better final solution.
> >
> > **Why SVD-LLM may be insufficient:** Methods like SVD-LLM overlook approximation-induced distribution shifts across layers. When the retained rank is small, approximation errors accumulate and the preserved component may fail to capture crucial information. Therefore, using SVD-LLM as a PEFT initialization can improve over *vanilla* LoRA in some settings, but it does not consistently match the benefits offered by our distribution-aware initialization.
> >
> > **Empirical evidence:** We tested on Vicuna-7B across a set of math and coding tasks. Table below reports the results (higher is better) for different initializations and rank settings; results are averaged across 3 seeds.
> >
> > | Rank | Method   | GSM8K | Math  | HumanEval | HumanEval+ |
> > |------|----------:|------:|------:|----------:|-----------:|
> > | 256  | Vanilla  | 0.293 | 0.034 | 0.177     | 0.146      |
> > |      | PiSSA    | 0.320 | 0.042 | 0.189     | 0.171      |
> > |      | SVD-LLM  | 0.321 | 0.041 | 0.189     | 0.165      |
> > |      | Ours     | **0.384** | **0.046** | **0.213** | **0.183** |
> > | 128  | Vanilla  | 0.306 | 0.036 | 0.159     | 0.128      |
> > |      | PiSSA    | 0.331 | 0.043 | 0.207     | 0.183      |
> > |      | SVD-LLM  | 0.379 | 0.049 | 0.171     | 0.146      |
> > |      | Ours     | **0.384** | **0.049** | **0.220** | **0.195** |
> >
> > These results show that: (1) SVD-LLM can outperform vanilla LoRA, (2) however SVD-LLM is not consistently better than PiSSA, and (3) our distribution-aware initialization yields the most consistent improvements across tasks and rank settings. In summary, both intuition and empirical results support that distribution-aware low-rank approximation provides a more effective initialization for PEFT than naive SVD-based methods. We also add these clarifications and the corresponding discussion in the revised manuscript.

---

### Author Response · Authors · 2025-11-22
**Update Manuscript**

We sincerely thank the reviewers for their careful reading and constructive comments. We are also encouraged that the reviewers found our paper to be well written (xHtd, E5cT), provides clear and well-explained theoretical analysis (E5cT, 8dzz), and demonstrates effective empirical results (xHtd, 8dzz). We uploaded a revised version of our paper and marked the major modifications in blue for visibility. In short,


1. We conducted additional experiments on Qwen2.5-7B, 14B, and 32B to verify the scalability of our method to larger models. (Table 1)
2. We added quantitative discussions on throughput and GPU memory usage under different compression ratios, output lengths, and batch sizes. (Appendix G)
3. We applied our method to large vision-language models, demonstrating its effectiveness in multi-modal scenarios. (Appendix H)
4. We reorganized the experimental section to move discussions of post-compression fine-tuning and comparisons with pruning-based methods into the main text for easier reader comprehension. (Subsection 4.2.3 & Subsection 4.2.4)
5. We added visualization of the Lipschitz constants of different modules in LLMs and identified consistent trends across models. (Appendix E)
6. We supplemented details on experimental settings to avoid potential reader confusion. (Subsection 4.1)

Thank you all again for your valuable and insightful suggestions. Please let us know if you have additional questions or ideas for improvement.

Kind regards, Authors

---

### Meta-Review · Area_Chair_noLZ · 2026-01-01

**Summary:**

* The novelty of the proposed work feels thin. Core contribution is whitening + SVD. Distribution-aware/dynami framing still doesn't clearly separate it from prior whitening-/activation-aware low-rank methods.
* Table 1 compression often yields unusable PPL. The story relies on post-FT recovery, which weakens the claimed standalone compression contribution.
* Added calibration/whitening overhead and workflow complexity aren't convincingly justified in contrast to simpler compression methods based on pruning/quantization/standard SVD baselines. This matters for real deployments.

**Reviewer Concerns:**

Reviewer xHtd
* [Addressed] Concern that Table 1 PPL is too high:  rebuttal explains it's no recovery by design and adds lightweight post-FT results. PEFT ablation: rebuttal provides comparative table.
* [Partially addressed] Novelty: rebuttal reframes contribution as distribution-shift / error-propagation analysis + dynamic whitening, but this is unlikely to change the reviewer's contribution score.

Reviewer E5cT
* [Addressed] Missing efficiency/overhead: rebuttal adds compression-time + throughput/memory. Scalability: rebuttal adds Qwen2.5-14B/32B.
* [Partially addressed] Novelty: reviewer explicitly acknowledges the explanation but still views the core technique as limited novelty and keeps the rating.

Reviewer 8dzz
[Addressed] All questions clarified. Reviewer is clearly positive and recommends accept.

Reviewer i2oL
* [Not addressed] Novelty: although a normalization baseline is added, the reviewer’s deeper concern about incrementality and positioning relative to prior whitening-based compression likely remains.

**Reviewer Scores:**

* Reviewer xHtd: May have raised the score slightly --> 2 or 4 (from 2)
* Reviewer E5cT: Would have kept the score --> 6 (from 6)
* Reviewer 8dzz: Would have kept the score --> 8 (from 8)
* Reviewer i2oL: Would have kept the score --> 2 (from 2)

The main issue is a lack of novel ideas, as noted by 3 reviewers (even by one positive reviewer).

---

### Decision · Program_Chairs · 2026-01-26

Reject